# Modeling the dynamics of *Plasmodium falciparum* gametocytes in humans during malaria infection

**Pengxing Cao**[1]*, **Katharine A Collins**[2,3], **Sophie Zaloumis**[4],
**Thanaporn Wattanakul**[5,6], **Joel Tarning**[5,6], **Julie A Simpson**[4], **James McCarthy**[3],
**James M McCaw**[1,4,7]*

[1]School of Mathematics and Statistics, University of Melbourne, Melbourne, Australia; [2]Department of Medical Microbiology, Radboud University Medical Center, Nijmegen, Netherlands; [3]QIMR Berghofer Medical Research Institute, Brisbane, Australia; [4]Centre for Epidemiology and Biostatistics, Melbourne School of Population and Global Health, University of Melbourne, Melbourne, Australia; [5]Mahidol-Oxford Tropical Medicine Research Unit, Faculty of Tropical Medicine, Mahidol University, Bangkok, Thailand; [6]Centre for Tropical Medicine and Global Health, Nuffield Department of Medicine, University of Oxford, Oxford, United Kingdom; [7]Epidemiology, Peter Doherty Institute for Infection and Immunity, Parkville, Australia

**Abstract** Renewed efforts to eliminate malaria have highlighted the potential to interrupt human-to-mosquito transmission — a process mediated by gametocyte kinetics in human hosts. Here we study the in vivo dynamics of *Plasmodium falciparum* gametocytes by establishing a framework which incorporates improved measurements of parasitemia, a novel gametocyte dynamics model and model fitting using Bayesian hierarchical inference. We found that the model provides an excellent fit to the clinical data from 17 volunteers infected with *P. falciparum* (3D7 strain) and reliably predicts observed gametocytemia. We estimated the sexual commitment rate and gametocyte sequestration time to be 0.54% (95% credible interval: 0.30–1.00%) per asexual replication cycle and 8.39 (6.54–10.59) days respectively. We used the data-calibrated model to investigate human-to-mosquito transmissibility, providing a method to link within-human host infection kinetics to epidemiological-scale infection and transmission patterns.
DOI: https://doi.org/10.7554/eLife.49058.001

*For correspondence:
pengxing.cao@unimelb.edu.au
(PC);
jamesm@unimelb.edu.au (JMMC)

**Competing interests:** The authors declare that no competing interests exist.

## Introduction

Malaria is a mosquito-borne parasitic disease caused by protozoan parasites of the *Plasmodium* genus. It is estimated to have caused approximately 219 million new cases and 435,000 deaths in 2017, primarily due to *Plasmodium falciparum* (*The World Health Organization, 2018*). New tools will be required to achieve the ambitious goal of malaria elimination. Among the tools proposed are novel interventions to block transmission from human hosts to vector mosquitoes (*The malERA Refresh Consultative Panel on Tools for Malaria Elimination, 2017*). *P. falciparum* malaria is transmitted from humans to the mosquito when terminally differentiated male and female sexual-stages of the parasite, called gametocytes, are taken up by female *Anopheles* mosquito during a blood meal (*Bousema and Drakeley, 2011*; *Josling and Llinás, 2015*). The level of gametocytes in the blood, often referred to as gametocytemia, is highly associated with the probability of human-to-mosquito transmission (*Bradley et al., 2018*; *Churcher et al., 2013*). Gametocyte levels below a

certain threshold (i.e., <1000 per mL blood) minimize the probability that a mosquito will take up both a male and female gametocyte during a blood-meal, which is necessary to propagate infection (*Collins et al., 2018*). An accurate understanding of the kinetics of gametocyte development in the human host is essential to predict the probability of transmission. A mathematical model that accurately captures the processes that give rise to observed gametocyte kinetics would be an important predictive tool to facilitate the design and evaluation of effective intervention strategies.

There is significant uncertainty surrounding fundamental aspects of *P. falciparum* gametocyte dynamics in humans. Parameters such as how many gametocytes are produced during each asexual replication cycle, the period of time in which early gametocytes disappear from the circulation before mature gametocytes appear (referred to as sequestration), and the period in which gametocytes circulate are poorly quantified. These gaps in understanding are due to a range of technical and logistic limitations. The first is the relatively poor sensitivity of the standard diagnostic test, namely microscopic examination of blood-films. Previous in vivo estimates of gametocyte kinetic parameters have been primarily based on historical data from neurosyphilis patients who were treated with so-called malariotherapy (*Diebner et al., 2000*; *Eichner et al., 2001*). In these studies, the limit of quantification was approximately $10^4$ parasites/mL blood, at least two orders of magnitude higher than that of current quantitative PCR (qPCR) assays (*Rockett et al., 2011*). This high limit of quantification prevents an accurate estimation of onset of emergence of both asexual parasites and mature gametocytes in peripheral blood. The second limitation is that the available estimates of gametocyte dynamics parameters based on in vitro cultures (*Filarsky et al., 2018*; *Gebru et al., 2017*) may not be applicable to natural infection with *P. falciparum* gametocytes due to in vitro conditions that may not mimic the human host (*Bousema and Drakeley, 2011*).

Recent advances in experimental medicine using volunteer infection studies (VIS), otherwise known as controlled human malaria infection (CHMI) studies (*Coffeng et al., 2017*), allow prospective study design and data collection with the explicit aim of collecting in vivo data (*McCarthy et al., 2011*), in particular an improved quantification of *P. falciparum* gametocyte kinetics by qPCR applied in a novel VIS (*Collins et al., 2018*). Furthermore, the models and fitting methods used in the neurosyphilis patient studies have been superseded for parameter estimation by increasingly sophisticated within-host models (*Khoury et al., 2018*) and improvements in computational algorithms for Bayesian statistical inference (*Piray et al., 2019*). Therefore, there is an emerging opportunity to improve our quantitative understanding of the dynamics of *P. falciparum* gametocytes in human hosts by combining the novel VIS data and advanced modeling approaches.

In this study, we developed a novel mathematical model of gametocyte dynamics, fitted the model to the VIS data and estimated the gametocyte dynamics parameters using a Bayesian hierarchical inference method. We demonstrate that the data-calibrated model can reliably predict the time-course of gametocytemia and thus should form an essential part of modeling studies of malaria transmission.

## Results

### Model fitting and validation

The outcome variable used in model fitting was the total parasitemia (total circulating asexual parasites and gametocytes per mL blood measured using qPCR) collected from a previously published VIS (*Collins et al., 2018*), with other measurements from the same study, such as the asexual parasitemia (circulating asexual parasites per mL blood) and gametocytemia (circulating female and male gametocytes per mL blood), used to validate model predictions.

The results of fitting the mathematical model to total parasitemia data for all 17 volunteers are given in *Figure 1* where 12 of 17 volunteers experienced recrudescence. The median of posterior predictions and 95% prediction interval (PI) were computed from 5000 model simulations based on 5000 samples from the posterior parameter distribution (see Materials and methods). The results show that the predicted total parasitemia (median and 95% PI) is able to accurately capture the trends of the data through the (visual) posterior predictive check. Furthermore, the narrow 95% PI indicates a low level of uncertainty in predicted total parasitemia.

Having calibrated the model against total parasitemia, the 5000 posterior parameter sets were used to calculate the median of posterior predictions and 95% PI of the asexual parasitemia and

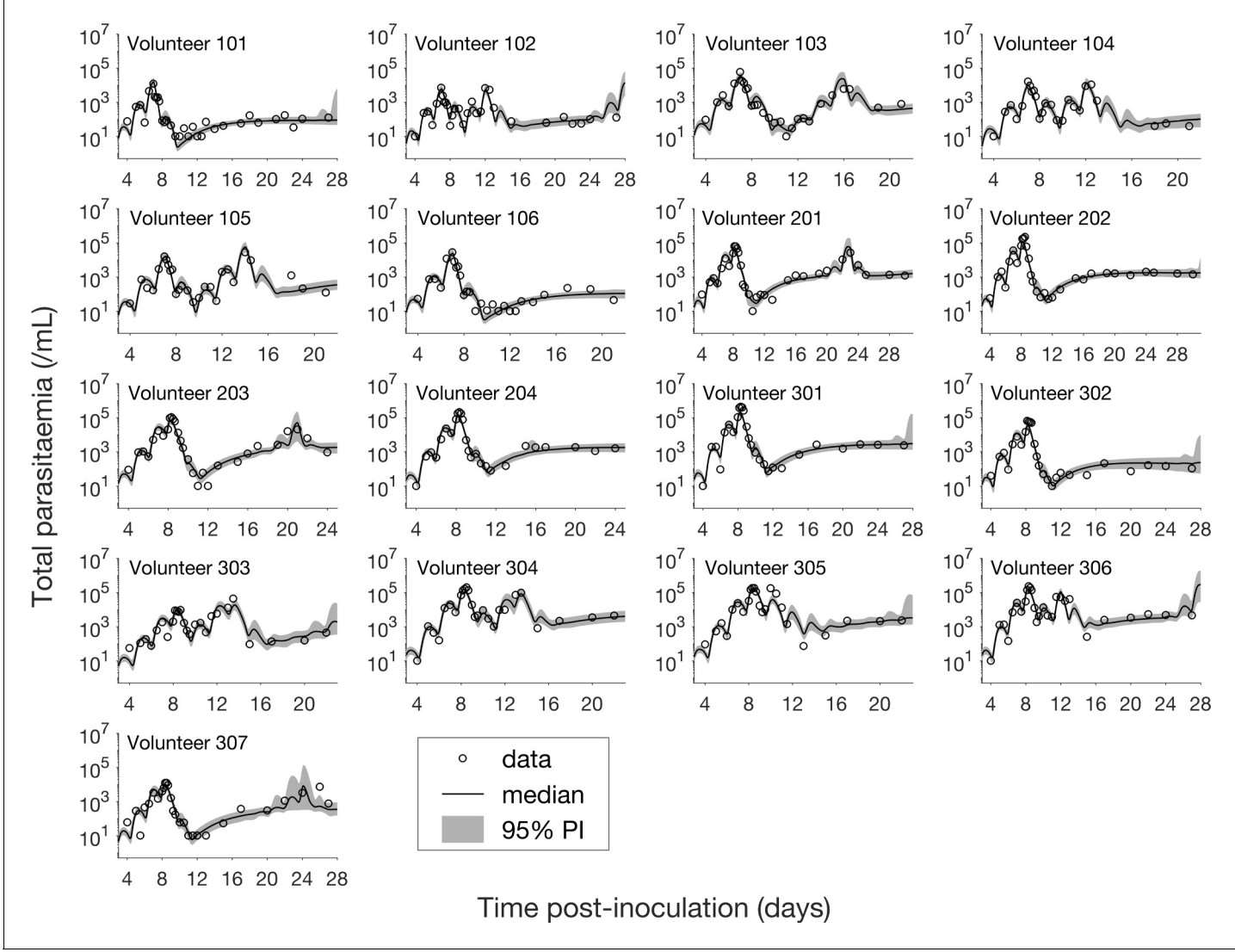

**Figure 1.** Results of model fitting for all 17 volunteers. Data are presented by circles. The median of posterior predictions (solid line) and 95% prediction interval (PI, shaded area) are generated by 5000 model simulations based on 5000 samples from the posterior parameter distribution as described in the Materials and methods. The histograms showing the posterior distributions of population mean and standard deviation hyperparameters are given in *Figure 1—figure supplements 1* and *2*. The posterior distribution of each model parameter (see the Materials and methods) for individual volunteers are given in *Figure 1—figure supplements 3–14* Posterior distributions for some biological parameters are given in *Figure 1—figure supplement 15*, which are generated based on the posterior samples of population mean parameters (see the Materials and methods) and will be used to support the results in *Table 1* shown later. The source data and computer code with instructions of implementation to generate *Figure 1* and *Figure 1—figure supplements 1–15* are fully publicly available at https://doi.org/10.26188/5cde4c26c8201.
DOI: https://doi.org/10.7554/eLife.49058.002

The following figure supplements are available for figure 1:

**Figure supplement 1.** Marginal posterior distributions for the 12 population mean parameters (hyperparameters).
DOI: https://doi.org/10.7554/eLife.49058.003

**Figure supplement 2.** Marginal posterior distributions for the 12 population SD parameters (hyperparameters).
DOI: https://doi.org/10.7554/eLife.49058.004

**Figure supplement 3.** The marginal posterior distributions of the individual parameter of $P_{init}$ (inoculation size) for all 17 volunteers.
DOI: https://doi.org/10.7554/eLife.49058.005

**Figure supplement 4.** The marginal posterior distributions of the individual parameter of $\mu$ (mean of the initial parasite age distribution) for all 17 volunteers.
DOI: https://doi.org/10.7554/eLife.49058.006

*Figure 1 continued on next page*

*Figure 1 continued*

**Figure supplement 5.** The marginal posterior distributions of the individual parameter of $\sigma$ (Standard deviation of the initial parasite age distribution) for all 17 volunteers.
DOI: https://doi.org/10.7554/eLife.49058.007

**Figure supplement 6.** The marginal posterior distributions of the individual parameter of $r_P$ (parasite replication rate) for all 17 volunteers.
DOI: https://doi.org/10.7554/eLife.49058.008

**Figure supplement 7.** The marginal posterior distributions of the individual parameter of $k_{max}$ (maximum rate of parasite killing by PQP) for all 17 volunteers.
DOI: https://doi.org/10.7554/eLife.49058.009

**Figure supplement 8.** The marginal posterior distributions of the individual parameter of $EC_{50}$ (half-maximum effective PQP concentration) for all 17 volunteers.
DOI: https://doi.org/10.7554/eLife.49058.010

**Figure supplement 9.** The marginal posterior distributions of the individual parameter of $\gamma$ (Hill coefficient for PQP) for all 17 volunteers.
DOI: https://doi.org/10.7554/eLife.49058.011

**Figure supplement 10.** The marginal posterior distributions of the individual parameter of $f$ (sexual commitment rate; not converted to percentage) for all 17 volunteers.
DOI: https://doi.org/10.7554/eLife.49058.012

**Figure supplement 11.** The marginal posterior distributions of the individual parameter of $\delta_P$ (death rate of asexual and sexual parasites) for all 17 volunteers.
DOI: https://doi.org/10.7554/eLife.49058.013

**Figure supplement 12.** The marginal posterior distributions of the individual parameter of $m$ (maturation rate of gametocytes) for all 17 volunteers.
DOI: https://doi.org/10.7554/eLife.49058.014

**Figure supplement 13.** The marginal posterior distributions of the individual parameter of $\delta_G$ (death rate of sequestered gametocytes) for all 17 volunteers.
DOI: https://doi.org/10.7554/eLife.49058.015

**Figure supplement 14.** The marginal posterior distributions of the individual parameter of $\delta_{Gm}$ (death rate of circulating gametocytes) for all 17 volunteers.
DOI: https://doi.org/10.7554/eLife.49058.016

**Figure supplement 15.** Marginal posterior distributions of some key biological parameters.
DOI: https://doi.org/10.7554/eLife.49058.017

**Table 1.** Estimates of some key biological parameters and comparison with the literature.
The estimates of the biological parameters (middle column) are shown as the median and 95% credible interval (CI) of the marginal posterior parameter distribution (*Figure 1—figure supplement 15*). Estimates from the literature (third column) are shown in the format of either 'mean estimate (95% confidence interval)' or 'mean estimate [minimum – maximum estimate]' or simply 'a low estimate – a high estimate'. Note some quoted estimates are from either in vivo or in vitro studies of *P. falciparum*. The source data and computer code with instructions of implementation to generate our model estimates (middle column) in *Table 1* are fully publicly available at https://doi.org/10.26188/5cde4c26c8201.

| Biological parameters (unit) | Median estimate (95% CI) | Estimates in the literature |
|---|---|---|
| Sexual commitment rate (%/asexual replication cycle) | 0.54 (0.30–1.00) | 11 (6.2–15.8) (*Filarsky et al., 2018*) (in vitro) 0.64 [0.027–13.5] (*Eichner et al., 2001*) (in vivo) |
| Gametocyte sequestration time (days) | 8.39 (6.54–10.59) | 7.4 [4 – 12] (*Eichner et al., 2001*) (in vivo) |
| Circulating gametocyte lifespan (days) | 63.5 (12.7–1513.9) | 16–32 (*Gebru et al., 2017*) (in vitro) 6.4 [1.3–22.2] (*Eichner et al., 2001*) (in vivo) |
| Parasite multiplication factor (per asexual replication cycle) | 21.8 (17.6–26.9) | 10–33 (*Wockner et al., 2017*) (in vivo) 16.4 (15.1–17.8)[a] (in vivo) |

[a] JS McCarthy, personal communication, May 2019.
DOI: https://doi.org/10.7554/eLife.49058.018

gametocytemia versus time profiles. Model predictions of the asexual parasitemia and gametocytemia for all 17 volunteers are shown in *Figure 2* and *Figure 3* respectively (curves: median prediction; shaded areas: 95% PI) and are compared to the asexual parasitemia and gametocytemia data (circles). We emphasize that this was not a fitting exercise, rather an independent validation of the calibrated model.

For the majority of asexual parasitemia data the model predictions (median and 95% PI) can faithfully capture the trends of the data (*Figure 2*), in particular the accurate predictions for both the recrudescent case where a portion of asexual parasitemia data diverge from the total parasitemia measurement (e.g., Volunteer 103, 104, 105, 201, 203, 304 and 307) and the non-recrudescent case where the posterior-median prediction curve (solid red curve) lies below the limit of detection (one asexual parasite/mL) (e.g., Volunteer 202, 301 and 302). However, there are some discrepant observations. The model under-predicts (Volunteer 204) or over-predicts (Volunteer 303, 305 and 306) a

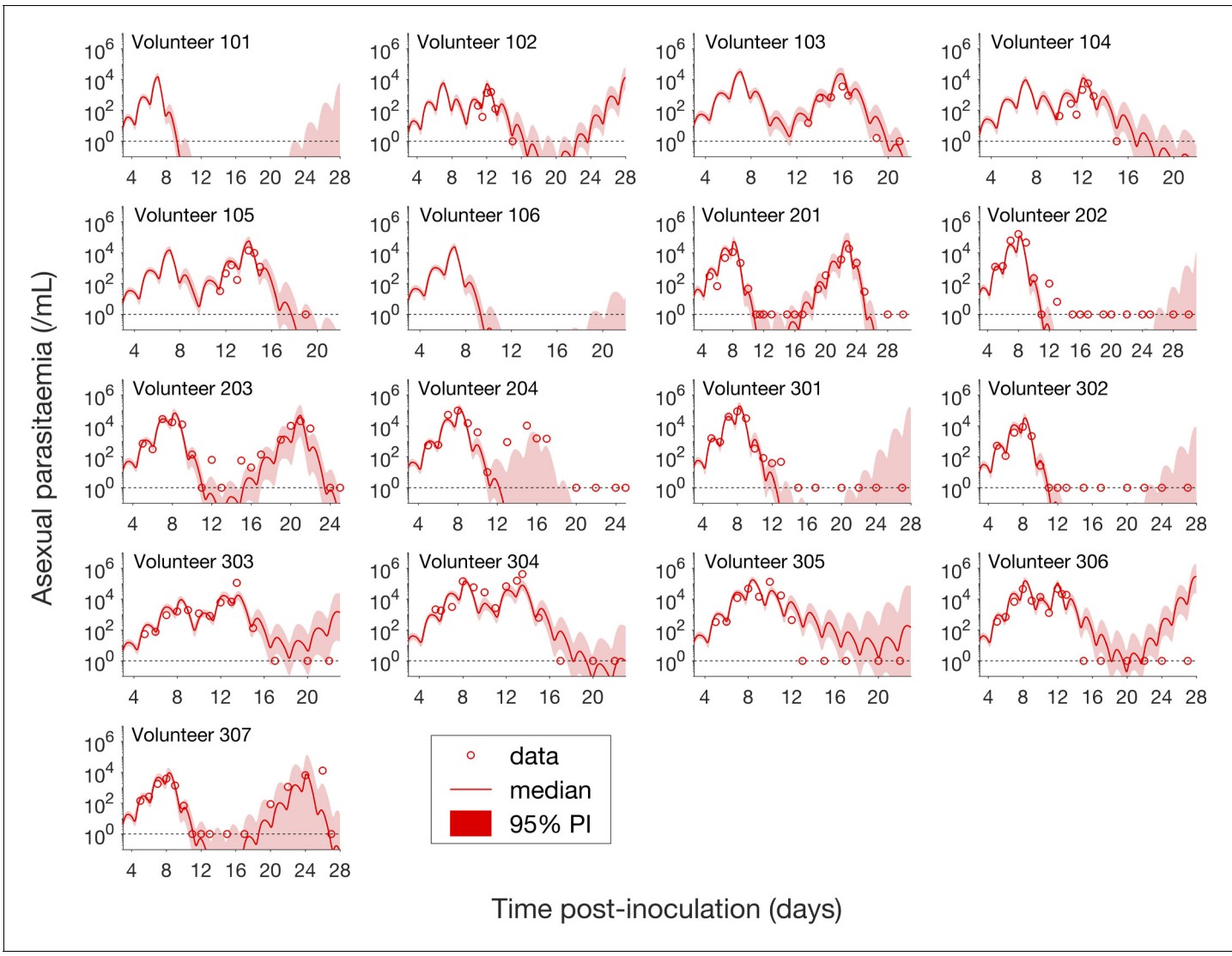

**Figure 2.** Comparison of model predictions and clinical data for the asexual parasitemia for all 17 volunteers. Data are presented by circles. The median of posterior predictions (solid curve) and 95% PI (shaded area) are generated by 5000 model simulations based on 5000 samples from the posterior parameter distribution as described in the Materials and methods. The data points with one parasite/mL (i.e., those points which lie on the dotted line) indicate measurements for which no parasites were detected. No data are available for Volunteer 101 and 106 to validate the model predictions. The source data and computer code with instructions of implementation to generate *Figure 2* are fully publicly available at https://doi.org/10.26188/5cde4c26c8201.
DOI: https://doi.org/10.7554/eLife.49058.019

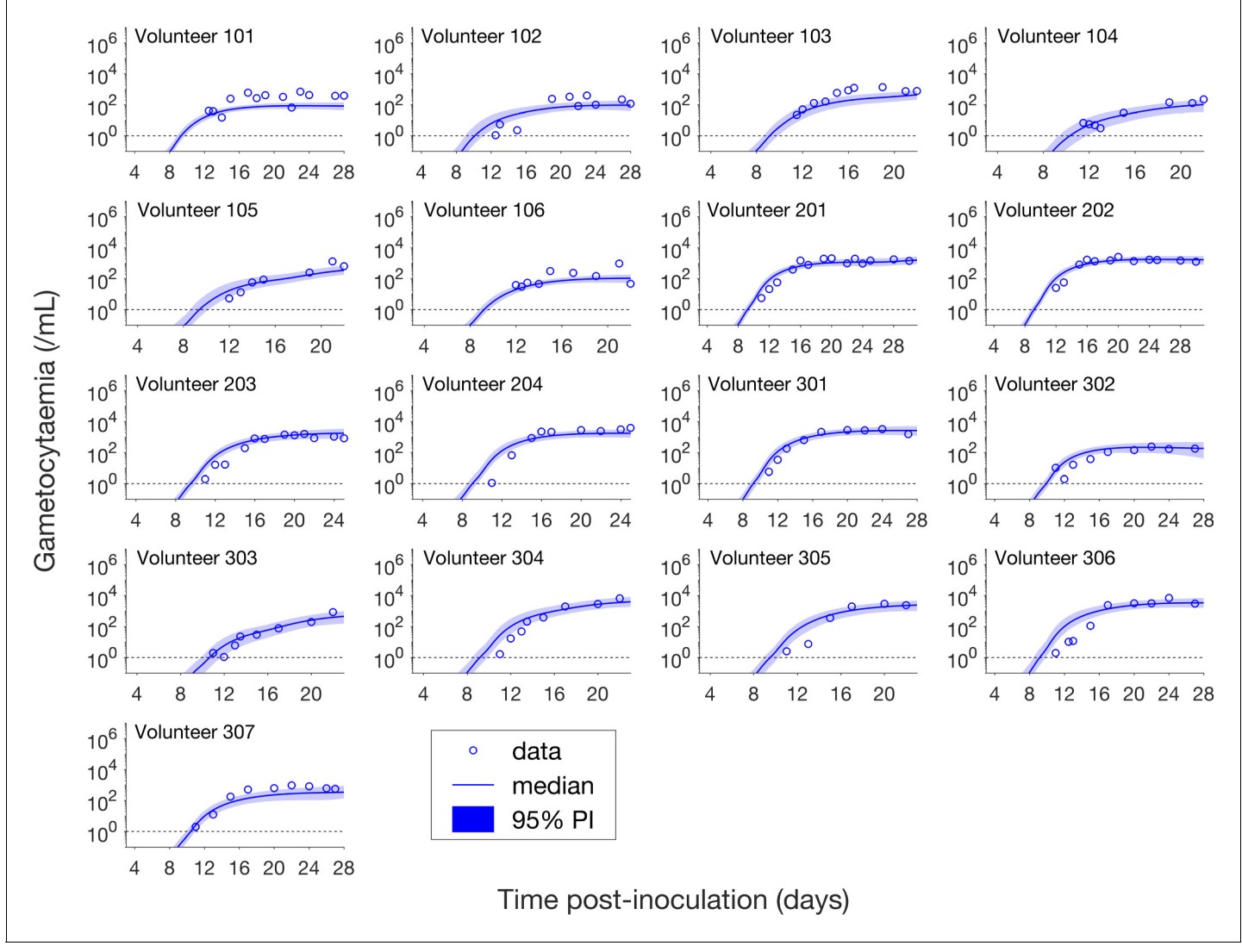

**Figure 3.** Comparison of model predictions and clinical data for the gametocytemia for all 17 volunteers. Data are presented by circles. The median of posterior predictions (solid curve) and 95% PI (shaded area) are generated by 5000 model simulations based on 5000 samples from the posterior parameter distribution as described in the Materials and methods. The data points with one parasite/mL (i.e. those points which lie on the dotted line) indicate measurements for which no parasites were detected. The source data and computer code with instructions of implementation to generate *Figure 3* are fully publicly available at https://doi.org/10.26188/5cde4c26c8201.
DOI: https://doi.org/10.7554/eLife.49058.020

portion of the asexual parasitemia data. Furthermore, for some volunteers such as 202, 301 and 302, the 95% PI (red shaded area) extends into the detectable region again after the asexual parasitemia reaches below the detection limit, indicating that there was a small chance that the patients may have suffered a recrudescence during the observation period (of course, they did not) or after the observation period (although this predication cannot be evaluated because artemisinin combination therapy was given immediately after the period).

*Figure 3* shows the data and model predictions for the gametocytemia. Despite some discrepant observations for asexual parasitemia in *Figure 2*, we found that the model predictions of gametocytemia were able to capture the trends and levels of the gametocytemia data for all 17 volunteers.

### Estimation of gametocyte dynamics parameters

The model calibration process provided the joint posterior density for the model parameters, which were used to estimate some key biological parameters governing the dynamics of *P. falciparum*

gametocytes (detailed in the Materials and methods). As shown in *Table 1*, the sexual commitment rate — the percentage of asexual parasites entering sexual development during each asexual replication cycle — is estimated to be 0.54%/asexual replication cycle (95% credible interval (CI): 0.30–1.00%). This in vivo estimate of 0.54%/asexual replication cycle is much lower than 11%/asexual replication cycle that was estimated from in vitro data (*Filarsky et al., 2018*). The proportion of committed asexual parasites that survive to become mature gametocytes, calculated by discounting the sexual commitment rate by the probability of survival from the immature (stages I–IV) to mature (stage V) gametocyte life-stage, is 0.52%/asexual replication cycle (95% CI: 0.28–0.95%). The gametocyte sequestration time is the average time that immature gametocytes (stages I–IV) cannot be observed in the peripheral circulation. They re-emerge in the peripheral circulation as mature gametocytes (stage V). It was estimated to be 8.39 days (95% CI: 6.54–10.59 days). The estimate for the circulating gametocyte lifespan is 63.5 days, with a broad 95% CI (12.7–1513.9 days) resulting from the long-tailed posterior distribution (*Figure 1—figure supplement 15*) and is much longer than the previous in vitro estimate of 16–32 days (*Gebru et al., 2017*) (note that our lower bound of the 95% CI is lower than the in vitro estimated range). The wide estimate for the circulating gametocyte lifespan, and in particular the high upper bound of the 95% CI, is due to the limited observation time in the VIS which does not enable the lifespan to be accurately determined (explored in more detail in the Discussion).

As shown in *Table 1*, there are similarities in parameter estimates for *P. falciparum* between our analysis and the analysis of historical neurosyphilis patient data (*Eichner et al., 2001*). We found that they exhibited similar in vivo sexual commitment rate (median 0.54%/asexual replication cycle vs. mean 0.64%/asexual replication cycle with overlapping plausible ranges) and gametocyte sequestration time (median 8.39 days vs. mean 7.4 days with overlapping plausible ranges).

Finally, we provided an estimate for the parasite multiplication factor which is the average number of infected red blood cells produced by a single infected red blood cells after one replication cycle. The parasite multiplication factor is an important parameter that quantifies the net growth of asexual parasites and thus influences the rate of gametocyte generation. Our posterior-median estimate is 21.8 parasites per asexual replication cycle (95% CI: 17.6–26.9), consistent with previous estimates which lie in the range 10–33 (*Wockner et al., 2017*), and a bit larger than an updated estimate calculated from a pooled analysis of parasite counts from 177 volunteers infected with the same *P. falciparum* strain using a statistical model (16.4 parasites per asexual replication cycle) (JS McCarthy, personal communication, May 2019).

## Predicting the impact of gametocyte kinetics on human-to-mosquito transmissibility

Having validated our mathematical model of asexual parasitemia and gametocyte dynamics, we were able to predict how the gametocyte dynamics parameters influence the transmissibility of *P. falciparum* malaria from humans to mosquitoes in various epidemiological scenarios. In particular, we focused on the early phase of infection where the innate immune response is minimal and treatment has not been administered (in order to avoid complications that our mathematical model was not designed to capture). Two scenarios were considered:

- Predicting the potential infectiousness of newly hospitalized clinical malaria cases for various values of sexual commitment rate and gametocyte sequestration time. In the model, gametocytemia was assumed to be a surrogate of the potential infectiousness. We further assumed that patients would seek hospital admission when their total parasitemia reached approximately $10^8$ parasites/mL. This choice was based on the microscopic measurements of the total parasitemia (i.e., asexual and sexual parasites) from a study of Cambodia and Thailand hospitalized malaria patients (*Saralamba et al., 2011*). As illustrated in *Figure 4A*, we simulated the model (for different sexual commitment rates and gametocyte sequestration times) and looked at the critical gametocytemia level (indicated by $G_c$) corresponding to the time when the total parasitemia (wave-like black curves) first reached $10^8$ parasites/mL (their associations are indicated by the dotted lines and arrows).

- Predicting the non-infectious period of malaria patients for various values of sexual commitment rate and gametocyte sequestration time. In the model, the non-infectious period was defined to be time from the inoculation of infected red blood cells to the time when the gametocytemia reached $10^3$ parasites/mL, which is a threshold below which human-to-mosquito

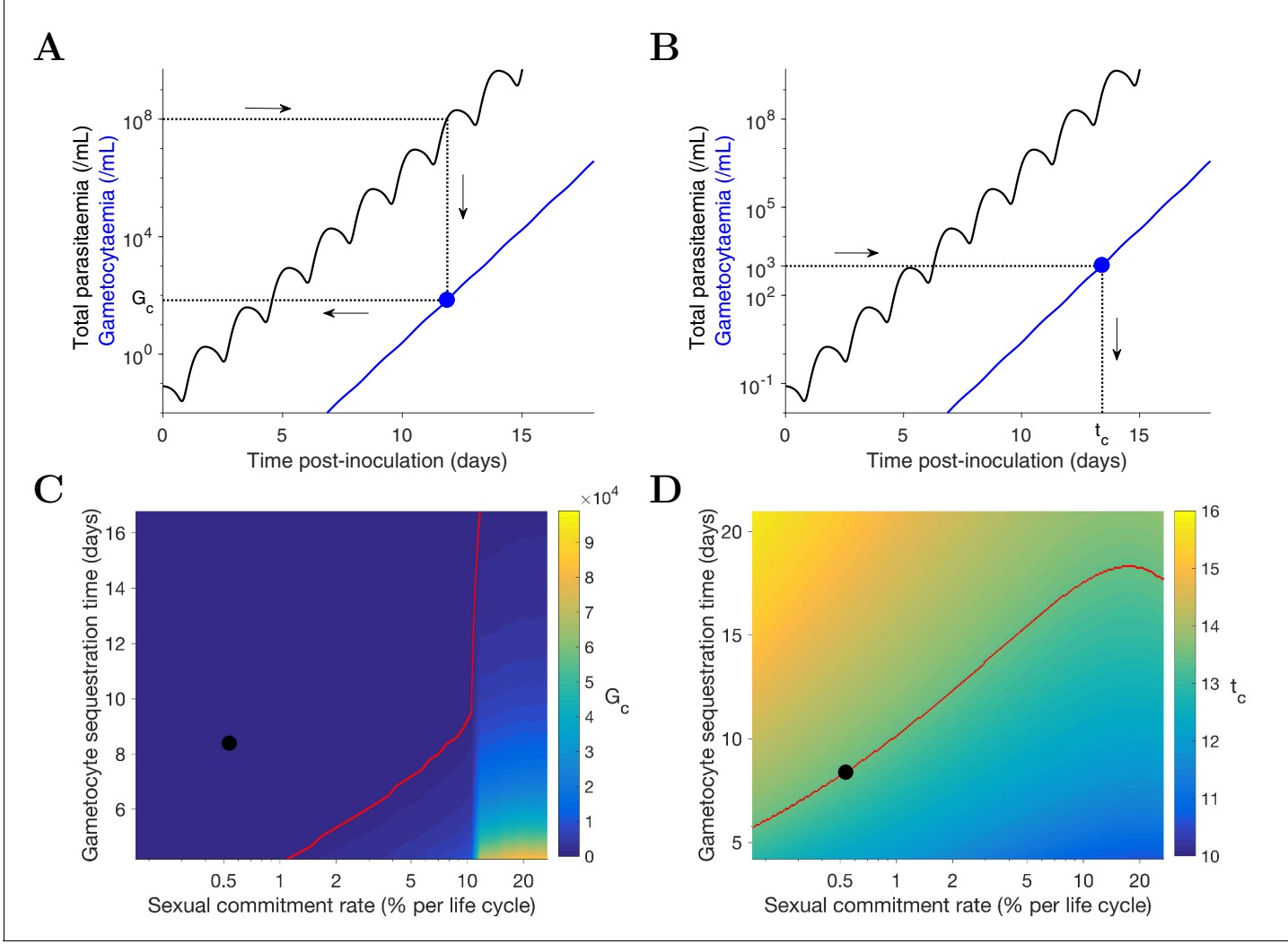

**Figure 4.** Simulation of two scenarios predicting the dependence of human-to-mosquito transmissibility on the sexual commitment rate and gametocyte sequestration time. (**A**) illustration of the first scenario: predicting the critical gametocytemia level (indicated by $G_c$) at the time when the total parasitemia reaches $10^8$ parasites/mL. (**B**) illustration of the second scenario: predicting the non-infectious period (indicated by $t_c$), which is defined to be time from inoculation of infected red blood cells to the time when the gametocytemia reaches $10^3$ parasites/mL (a threshold below which human-to-mosquito transmission was not observed [***Collins et al., 2018***]). (**C and D**) Heatmaps showing the dependence of the critical gametocytemia $G_c$ and the non-infectious period $t_c$ on the sexual commitment rate and gametocyte sequestration time. The black dots represent the value obtained by simulating the gametocyte dynamics model using the median estimates of the posterior samples of the population mean parameters as described in the Materials and methods. The red curve in C is the level curve for $G_c = 10^3$ parasites/mL. The red curve in D is the level curve for $t_c = 13.42$ days which is the non-infectious period obtained by model simulation using the posterior estimates of the population mean parameters. The source data and computer code with instruction of implementation to generate *Figure 4* are fully publicly available at https://doi.org/10.26188/5cde4c26c8201.
DOI: https://doi.org/10.7554/eLife.49058.021

transmission was not observed (***Collins et al., 2018***). Note that this non-infectious period does not include the latent period due to the liver stage which should be considered if the starting time were to be taken from time of mosquito bite. As illustrated in *Figure 4B*, we simulated the model (for different sexual commitment rates and gametocyte sequestration times) and identified the critical time (indicated by $t_c$) when the gametocytemia (blue curve) first reached $10^3$ parasites/mL (their associations are indicated by the dotted lines and arrows).

A higher sexual commitment rate or a lower gametocyte sequestration time leads to a higher gametocytemia ($G_c$) at the time of hospitalization (*Figure 4C*). The red curve in *Figure 4C* indicates the level curve of $10^3$ gametocytes/mL (i.e., the threshold for infectiousness as mentioned above) dividing the heatmap into two regions. To the left, $G_c$ is below $10^3$ gametocytes/mL, suggesting

clinical presentation precedes infectiousness, while to the right $G_c$ is above $10^3$ gametocytes/mL and the converse applies. The $G_c$ value obtained by model simulation using the median estimates of the population mean parameters (indicated by the black dot) is below $10^3$ gametocytes/mL, suggesting that newly hospitalized malaria patients are less likely to be infectious, and thus efforts to identify and treat infections in a timely manner may have a substantial impact in terms of reduced transmission potential. Note that patients from clinical observations of uncomplicated malaria in endemic settings may have higher gametocyte counts at the time of presentation than what our model predicts. For example one study from the TRACII clinical trial reported a range of 16–5120 gametocytes/µL, which is much higher than our prediction of below one gametocytes/µL (or $10^3$ gametocytes/mL) (*van der Pluijm et al., 2019*). One plausible explanation for the difference is that our model predicted a very fast rise in total parasitemia to $10^8$ parasites/mL while the rise in parasitemia among patients in endemic settings may be slower due to the effect of immunity on the parasite multiplication. Immunity was not considered in our model due to the design of our VIS where only malaria naïve volunteers were recruited.

*Figure 4D* reinforces the result in *Figure 4C* using the non-infectious period ($t_c$). As the sexual commitment rate increases or the gametocyte sequestration time decreases, $t_c$ decreases. However, for large values of the sexual commitment rate (e.g., >20%), we observed an increase in $t_c$ as the sexual commitment rate increases (see the top-right corner of ). This is because an increased sexual commitment rate leads to both a decrease in the rate of asexual parasite growth (due to a direct loss of asexual parasites as they convert to gametocytes) and an increase in the number of sexually committed parasites. For a very high sexual commitment rate, the impact of the former more than counterbalances that of the latter.

## Discussion

We have developed a novel mathematical model of gametocyte dynamics that combines an existing multi-state asexual cycle model with a new model for the development of gametocytes. Model parameters were estimated by fitting the model to data from 17 malaria-naïve volunteers inoculated with *P. falciparum*-infected red blood cells (3D7 strain). Compared to previous studies, our work is distinguished by three novel contributions: (1) the use of a prospectively planned clinical trial to collect more accurate quantitative data of parasite levels measured by qPCR; (2) the development of a novel dynamics mathematical model which allows for robust and biologically-informed extrapolation and hypothesis testing/scenario analysis; and (3) the use of a Bayesian hierarchical inference method for model calibration and parameter estimation.

For gametocyte kinetic parameters, we found that our in vivo estimate of the *P. falciparum* sexual commitment rate was similar to that found in the neurosyphilis patient data (*Eichner et al., 2001*) but was much smaller than previous in vitro estimates (*Table 1*). Importantly, our estimate follows directly from the structure of our mathematical model, and accounts for the fact that some early committed gametocytes may not complete development and thus not emerge in peripheral circulation as mature gametocytes. Novel VIS data using biomarkers specific to early sexual parasites (e.g. AP2-G [*Bancells et al., 2019*] and PfGEXP5 [*Tibúrcio et al., 2015*]) would enable a direct (statistical) estimate of the sexual commitment rate, providing an independent validation of our gametocyte dynamics model. Our in vivo estimate for the circulating gametocyte lifespan is imprecise (i.e., has a very wide credible interval) due to the lack of available data for gametocyte clearance (treatment was initiated before gametocyte were naturally cleared in the VIS study). *P. falciparum* data with gametocytemia measurements over a longer period of time to capture the natural decay of circulating gametocytes, would greatly improve these estimates.

We also predicted the effects of altered gametocyte kinetic parameters on the transmissibility from humans to mosquitoes, focusing on two scenarios: the infectiousness of newly hospitalized clinical malaria cases (i.e., the gametocytemia when total parasitemia first reaches a level typically seen upon hospitalization — $10^8$ parasites/mL in the model); and the non-infectious period of malaria patients (i.e., the time from the inoculation of infected red blood cells to the time when the gametocytemia reaches a minimal transmission threshold of $10^3$ parasites/mL in the model). We explored how the sexual commitment rate and gametocyte sequestration time influenced the gametocyte level and the non-infectious period. We would like to emphasize that human-to-mosquito transmissibility is determined by both the level of gametocytemia and the relationship between

gametocytemia and the probability of transmission per bite. A reliable prediction of the former is essential but not a sole determinant of transmissibility. Therefore, it is also important to improve our quantitative understanding of the probability of transmission per bite, which may be complicated by and also influenced by the densities and ratios of female and male gametocytes (*Bradley et al., 2018*; *Churcher et al., 2013*; *Da et al., 2015*).

Our study has some limitations. The gametocyte dynamics model, that has been shown to have sufficient complexity to reproduce the clinical observations, is still a rather coarse simplification of the actual biological processes. For example, the model does not assume an adaptive sexual commitment rate (*Schneider et al., 2018*), nor does it consider the mechanisms of sexual commitment (*Bancells et al., 2019*). Furthermore, the model assumes a constant gametocyte death rate but does not consider other non-constant formulations as have been previously proposed (*Diebner et al., 2000*). Another limitation is that we assumed a fixed duration for the asexual replication cycle of 42 hr, while previous work by our group suggests that the replication cycle may be altered by up to a few hours in response to antimalarial drugs (e.g., artemisinin [*Cao et al., 2017*; *Dogovski et al., 2015*]), though there is no evidence that piperaquine (which was administered in this VIS) has a similar effect.

In conclusion, we have developed a novel mathematical model of gametocyte dynamics, and demonstrated that it reliably predicts time series data of gametocytemia. The model provides a powerful predictive tool for informing the design of future volunteer infection studies which aim to test transmission-blocking interventions. Furthermore, the within human host transmission model can be incorporated into epidemiological-scale models to refine predictions of the impacts of various antimalarial treatments and transmission interventions.

## Materials and methods

### Study population and measurements

The data used in this modeling study are from a previously published VIS (*Collins et al., 2018*) where 17 malaria-naïve volunteers were inoculated with approximately 2800 *P. falciparum*-infected red blood cells (3D7 strain). The study was approved by the QIMR Berghofer Human Research Ethics Committee and registered with ClinicalTrials.gov (NCT02431637 and NCT02431650). The volunteers were treated with 480 mg piperaquine phosphate (PQP) on day 7 or 8 post-inoculation to attenuate asexual parasite growth and a second dose of 960 mg PQP was given to any volunteer for treatment of recrudescent asexual parasitemia. All volunteers received a course of artemether/lumefantrine and, if required, a single dose of primaquine (45 mg) to clear all parasites. Parasitemia in the volunteers was monitored approximately daily following inoculation, but with notable variability in the frequency of data collection at later times as described by *Collins et al. (2018)*.

Molecular analysis of parasite levels was carried out throughout the study. The total parasitemia was measured by 18S qPCR (total circulating asexual parasites and gametocytes per mL blood), asexual parasitemia was measured by SBP1 qRT-PCR (circulating asexual parasites per mL blood), and gametocytemia was measured by Pfs25 and PfMGET qRT-PCR (circulating female and male gametocytes per mL blood). Plasma concentrations of PQP were also determined at multiple time points after inoculation. Further details about the VIS are given in *Collins et al. (2018)*. It is important to note that the data used in model fitting is the total parasitemia (from the first measurement to the time before any treatment other than PQP) and the other data, that is asexual parasitemia and gametocytemia (also up to the time of treatment other than PQP) are used to validate the model.

### Gametocyte dynamics model

The mathematical model extends the published models of asexual parasite replication cycle (*Saralamba et al., 2011*; *Zaloumis et al., 2012*) by incorporating the development of gametocytes. The model is comprised of three parts describing three populations of parasites: asexual parasites ($P$), sexually committed parasites ($P_G$) and gametocytes ($G$). A schematic diagram of the development of those populations based on current knowledge (*Bancells et al., 2019*; *Filarsky et al., 2018*) is shown in *Figure 5*.

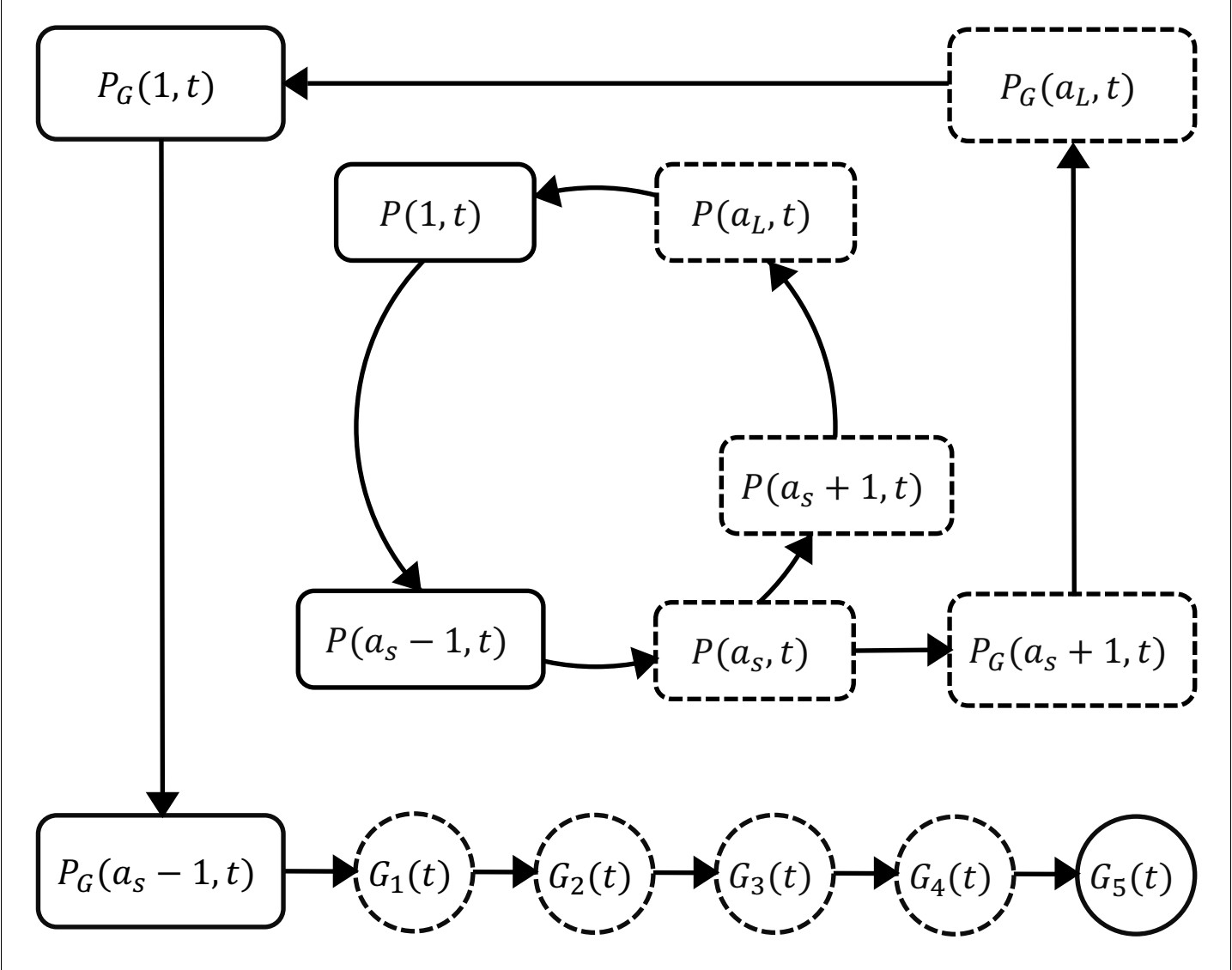

**Figure 5.** Schematic diagram showing the model compartments and transitions. The model is comprised of three parts describing three populations of parasites: asexual parasites ($P(a,t)$), sexually committed parasites ($P_G(a,t)$) and gametocytes ($G(t)$). P and $P_G$ are functions of asexual parasite age $a$ and time $t$. Square compartments in the inner loop represent the asexual parasite population which follows a cycle of maturation and replication every $a_L$ hours. Sexual commitment occurs from age $a_s$ and a fraction of asexual parasites become sexually committed (the bigger square compartments in the outer loop) and eventually enter the development of stage I–V gametocytes ($G_1$–$G_5$). The compartments with a dashed boundary are sequestered to tissues and thus not measurable in a blood smear. The notation for each compartment is consistent with those in the model equations and is explained in the main text.

DOI: https://doi.org/10.7554/eLife.49058.022

Asexual parasites develop and replicate in the red blood cells (RBCs) until cell rupture at the end of each replication cycle and the released free parasites (merozoites) can initiate new cycles of replication if they successfully invade susceptible RBCs. At the time of inoculation (i.e., $t = 0$ hours in the model), we define the inoculum size to be $P_{init}$ and assume the age distribution of inoculated parasites is Gaussian with mean μ and standard deviation $\sigma$. As time increments by one hour, the asexual parasites of age $a$ at time $t$ (denoted as $P(a,t)$) follow the iterative equation:

$$P(a,t) = \begin{cases} P(a-1,t-1)e^{-\overline{k_d}-\delta_P}, & a = 2, 3, ..., a_L \\ r_P P(a_L, t-1)e^{-\overline{k_d}-\delta_P}, & a = 1 \end{cases} \tag{1}$$

where $\overline{k_d}$ represents the average rate of asexual parasite killing by PQP and $\delta_P$ is the rate of asexual parasite death due to processes other than PQP. $\overline{k_d}$ is approximated by the average of $k_d(t-1)$ and $k_d(t)$ and $k_d(t) = k_{max}C(t)^{\gamma}/(C(t)^{\gamma}+EC_{50}^{\gamma})$ where $k_{max}$ is the maximum killing rate, $EC_{50}$ is the PQP concentration at which half maximum killing is achieved, and $\gamma$ is the Hill coefficient determining the curvature of the dose-response curve. $C(t)$ is the PQP concentration which is simulated by a pharmacokinetic model introduced below. $a_L$ is the length of each asexual replication cycle and $r_P$ is the parasite replication rate indicating the average number of newly infected RBCs attributable to the rupture of a single infected RBC. Note that we distinguish the parasite replication rate $r_P$ from the so-called parasite multiplication factor, the latter of which is a 'net replication rate' quantified by the (per cycle) increase in parasite numbers due to replication ($r_P$) and the decrease in parasite numbers due to death or sexual commitment. Sexual commitment is assumed to occur at the first age of the trophozoite stage (denoted to be $a_s$) and a fraction ($f$) of asexual parasites leave the asexual replication cycle and start sexual development in the next hour, which is modeled by

$$P(a_s+1,t) = (1-f)P(a_s,t-1)e^{-\overline{k_d}-\delta_P} \tag{2}$$

$$P_G(a_s+1,t) = fP(a_s,t-1)e^{-\overline{k_d}-\delta_P}. \tag{3}$$

The first equation describes the proportion of parasites remaining in the asexual replication cycle while the second equation describes the proportion of parasites becoming sexually committed parasites ($P_G$). According to *Figure 5*, the sexually committed parasites continue the rest of the replication cycle and a part of the next replication cycle (note that they appear indistinguishable from asexual parasites by microscopy) before becoming stage I gametocytes. The process is modeled by

$$P(a,t) = \begin{cases} P(a-1,t-1)e^{-\delta_P}, & a=2,\ 3,\ ...,a_L \text{ except } a=a_s \text{ and } a=a_s+1 \\ r_P P(a_L,t-1)e^{-\delta_P}, & a=1 \end{cases} \tag{4}$$

Note that we assumed in our model that PQP does not kill gametocytes. Our assumption was based on evidence from both in vitro and in vivo experiments that suggests that PQP has little activity against sexually committed parasites and gametocytes (*Collins et al., 2018*; *Pasay et al., 2016*; *Bolscher et al., 2015*), although we note there is some evidence that PQP might have activity against early-stage I/II gametocytes (*Adjalley et al., 2011*). The changes of the sequestered stage I– IV gametocytes ($G_1$–$G_4$) are governed by difference equations

$$G_1(t) = G_1(t-1)e^{-(m+\delta_G)} + \frac{P_G(a_s-1,t-1)e^{-\delta_P}\left(1-e^{-(m+\delta_G)}\right)}{m+\delta_G}, \tag{5}$$

$$G_2(t) = G_2(t-1)e^{-(m+\delta_G)} + \frac{mG_1(t-1)\left(1-e^{-(m+\delta_G)}\right)}{m+\delta_G}, \tag{6}$$

$$G_3(t) = G_3(t-1)e^{-(m+\delta_G)} + \frac{mG_2(t-1)\left(1-e^{-(m+\delta_G)}\right)}{m+\delta_G}, \tag{7}$$

$$G_4(t) = G_4(t-1)e^{-(m+\delta_G)} + \frac{mG_3(t-1)\left(1-e^{-(m+\delta_G)}\right)}{m+\delta_G}, \tag{8}$$

where m is the rate of gametocyte maturation and $\delta_G$ is the death rate of sequestered gametocytes. Stage V gametocytes are circulating in bloodstream (and therefore can be measured from the peripheral blood film) modeled by

$$G_5(t) = G_5(t-1)e^{-\delta_{Gm}} + \frac{mG_4(t-1)\left(1-e^{-\delta_{Gm}}\right)}{\delta_{Gm}}, \tag{9}$$

where $\delta_{Gm}$ is the death rate of mature circulating gametocytes.

The total parasitemia in the model is given by $\sum_{a=1}^{a=a_s-1}[P(a,t) + P_G(a,t)] + G_5(t)$, which was fitted to the VIS data. After model fitting, we simulated the asexual parasitemia $\sum_{a=1}^{a=a_s-1} P(a,t)$ and gametocytemia $G_5(t)$ and compared them with associated data for model validation. *Table 2* presents all the model parameters and their units and descriptions.

## Pharmacokinetic model of piperaquine (PQP)

In the within-host model, the killing rate $k_d(t)$ is determined by PQP concentration $C(t)$ which was simulated from a pharmacokinetic (PK) model introduced in this section. The PK model, provided by Thanaporn Wattanakul and Joel Tarning (Mahidol-Oxford Tropical Medicine Research Unit, Bangkok), is a three-compartment disposition model with two transit compartments for absorption (see the schematic diagram in *Figure 6*).

Based on *Figure 6*, the model is formulated to be a system of ordinary differential equations:

$$\frac{dD}{dt} = -k_T D, \tag{10}$$

$$\frac{dT_1}{dt} = k_T D - k_T T_1, \tag{11}$$

$$\frac{dT_2}{dt} = k_T T_1 - k_T T_2, \tag{12}$$

$$\frac{dC}{dt} = \frac{k_T T_2 + q_1 P_1 + q_2 P_2 - q_1 C - q_2 C - q_c C}{V_c}, \tag{13}$$

**Table 2.** Details of the gametocyte dynamics model parameters.

The table includes the unit, description and prior distribution for each model parameter. For the uniform prior distributions (U), the lower bounds are non-negative based on the definitions of the model parameters and the upper bounds for the prior distributions were chosen to be sufficiently wide in order to accommodate all biologically plausible values from the literature (*Zaloumis et al., 2012*). We assumed parasites younger than 25h are circulating and thus fix $a_s$ to be 25h. For 3D7 strain, the asexual replication cycle is approximately 39–45h (based on in vitro estimates [*Duffy and Avery, 2017*] and personal communication [JS McCarthy, personal communication, May 2019]) and we fix $a_L$ to be 42h.

| Parameter | Unit | Description | Prior distribution |
|---|---|---|---|
| $P_{init}$ | parasites/mL | inoculation size | U(0, 10) |
| $\mu$ | h | mean of the initial parasite age distribution | U(0, 35) |
| $\sigma$ | h | SD of the initial parasite age distribution | U(0, 20) |
| $r_P$ | (unitless) | parasite replication rate | U(0, 100) |
| $k_{max}$ | h$^{-1}$ | maximum rate of parasite killing by PQP | U(0, 1) |
| $EC_{50}$ | ng/mL | half-maximum effective PQP concentration | U(1, 100) |
| $\gamma$ | (unitless) | Hill coefficient for PQP | U(0, 20) |
| $f$ | (unitless) | the fraction of parasites entering sexual development per asexual replication cycle | U(0, 1) |
| $\delta_P$ | h$^{-1}$ | death rate of asexual and sexual parasites | U(0, 0.2) |
| $m$ | h$^{-1}$ | maturation rate of gametocytes | U(0, 0.1) |
| $\delta_G$ | h$^{-1}$ | death rate of sequestered gametocytes | U(0, 0.1) |
| $\delta_{Gm}$ | h$^{-1}$ | death rate of circulating gametocytes | U(0, 0.1) |
| $a_s$ | h | sequestration age of asexual parasites | fixed to be 25 |
| $a_L$ | h | length of life cycle of asexual parasites | fixed to be 42 |

SD: standard deviation; h: hour.

DOI: https://doi.org/10.7554/eLife.49058.023

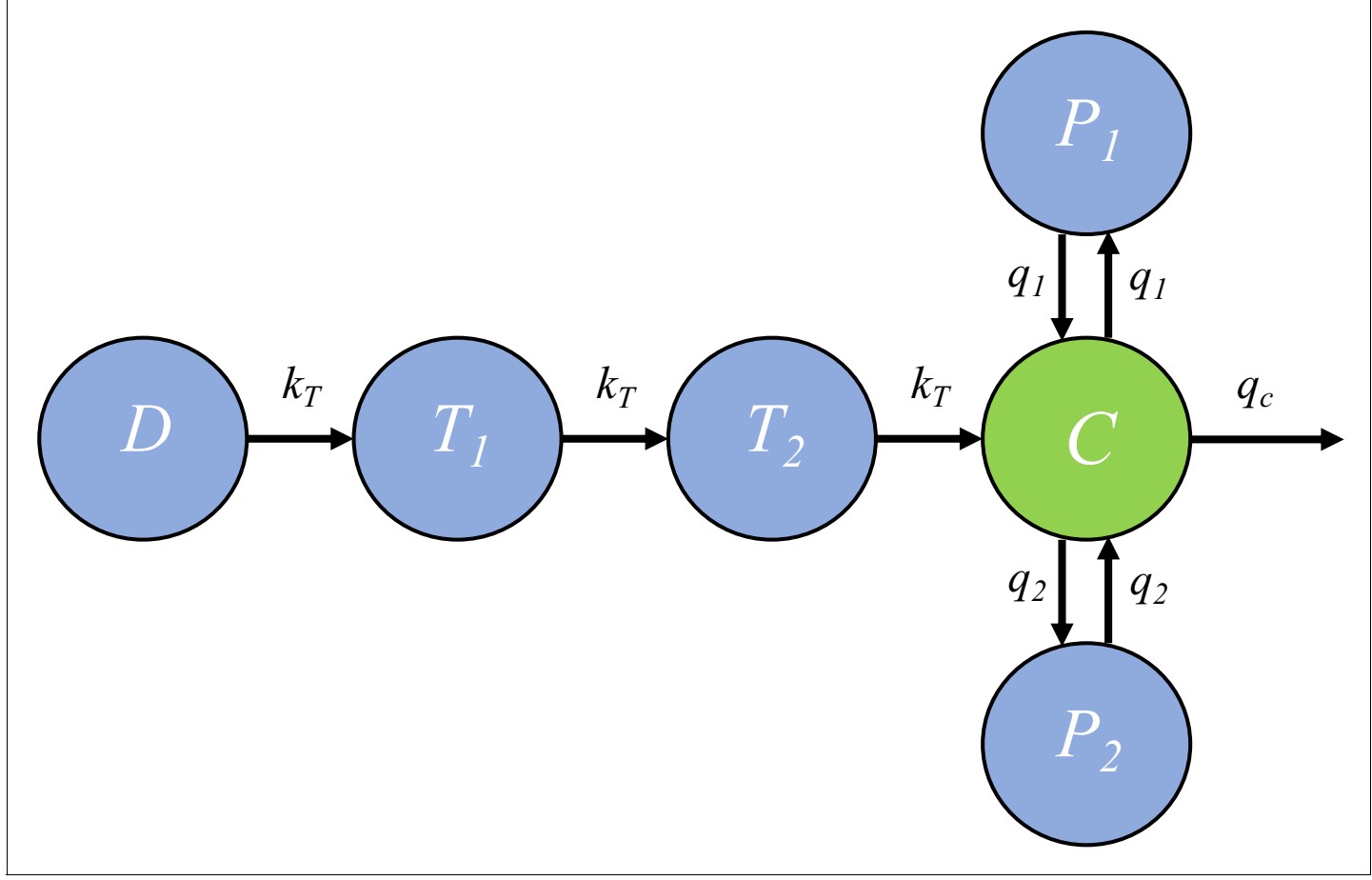

**Figure 6.** The pharmacokinetic model of piperaquine (PQP). The model is a three-compartment disposition model with two transit compartments for absorption. State $D$ represents the dose of PQP. $T_1$ and $T_2$ represent the two transit compartments. $C$ is the central compartment and PQP concentration in this compartment was measured (which are shown in **Figure 6—figure supplement 1**). $P_1$ and $P_2$ represent two peripheral compartments. $k_T$, $q_1$, $q_2$ and $q_c$ are the rates of flow into or out of compartments.
DOI: https://doi.org/10.7554/eLife.49058.024

The following figure supplement is available for figure 6:

**Figure supplement 1.** PK data and optimized PK curves (the 'fits') of piperaquine (PQP) concentration for all volunteers.
DOI: https://doi.org/10.7554/eLife.49058.025

$$\frac{dP_1}{dt} = \frac{q_1 C - q_1 P_1}{V_1} \,, \tag{14}$$

$$\frac{dP_2}{dt} = \frac{q_2 C - q_2 P_2}{V_2} \,, \tag{15}$$

where $k_T$ and $q$'s are rate constants as shown in **Figure 6** and $V_c$, $V_1$ and $V_2$ are the volume of distribution for the central compartment (in which PQP concentration is $C$), peripheral compartment 1 (in which PQP concentration is $P_1$) and peripheral compartment 2 (in which PQP concentration is $P_2$) respectively.

Under the sequential pharmacokinetic-pharmacodynamic (PK-PD) approach we have taken, a PQP concentration curve ($C(t)$) for each volunteer is a required input into the gametocyte dynamics model. The VIS, with its limited sampling of PQP for each volunteer, was not designed to provide this PQP concentration curve directly, so we used a PK model, informed by data from a previous VIS with rich sampling. We drew on an analysis of that previous VIS by Thanaporn Wattanakul and Joel

Tarning (unpublished data and estimates). Their analysis provides population-level PQP PK model parameter estimates.

We used MATLAB's (version 2016b; The MathWorks, Natick, MA) built-in least-squares optimizer *lsqcurvefit* (with the default setting) to optimize the PK curve for each volunteer in the VIS study. We applied the optimizer to each volunteer's (limited) PQP data, using the parameter estimates provided by Thanaporn Wattanakul and Joel Tarning as initial values. We applied some further model parameter constraints as specified in Appendix 1. This approach provided us with a data-informed PK curve for each volunteer in the VIS, sufficient for our primary purpose of studying the asexual and sexual parasite dynamics. Of note, Volunteers 202, 301, 302 or 307 had fewer PK data points than PK model parameters, preventing application of this optimization procedure. For these volunteers, their predicted PQP PK curve was derived using the population-level mean PK parameter from Wattanakul and Tarning's analysis. The MATLAB code (with detailed comments) is publicly available at https://doi.org/10.26188/5cde4c26c8201 The details of the initial conditions, starting point and constraints for the PK curve optimization procedure are provided in Appendix 1. The optimized PK curves and associated parameter values for all volunteers are provided in *Figure 6—figure supplement 1* and Appendix 1.

## Fitting the model to parasitemia data

We took a Bayesian hierarchical modeling approach (*Gelman et al., 2013*) to fit the gametocyte dynamics model to the data from all 17 volunteers. In detail, each volunteer has 12 model parameters (i.e., those in *Table 2* except $a_s$ and $a_L$; also called the individual parameters) and lower and upper bounds of the parameters are given in *Table 2*. If denoting the individual parameters to be $\theta_{ind}$ and lower and upper bounds to be $b_L$ and $b_U$ respectively, the following transformations are used to convert the bounded individual parameters to unbounded ones (denoted by $\varphi_{ind}$) in order to in order to improve computational efficiency (*Lesaffre et al., 2007*; *Stan Development Team, 2017*):

$$\varphi_{ind} = \ln\left(\frac{\theta_{ind} - b_L}{b_U - \theta_{ind}}\right),  \quad (16)$$

o$\varphi_{ind}$ beys a multivariate normal distribution $\mathfrak{N}(\varphi_{pop}, \Omega_{pop})$ where

$$\varphi_{pop} = \ln\left(\frac{\theta_{pop} - b_L}{b_U - \theta_{pop}}\right)  \quad (17)$$

and $\theta_{pop}$ is a vector containing 12 population mean parameters (hyperparameters) corresponding to the 12 gametocyte dynamics model parameters. $\Omega_{pop}$ is the covariance matrix. For more efficient sampling process, $\varphi_{ind} \sim \mathfrak{N}(\varphi_{pop}, \Omega_{pop})$ was reparameterised to a non-centerd form $\varphi_{ind} = \varphi_{pop} + \omega_{pop}L\eta$, where $\omega_{pop}$ is the diagonal standard deviation (SD) matrix whose diagonal elements are the 12 population SD parameters (hyperparameters); $L$ is the lower Cholesky factor of the correlation matrix; $\eta$ obeys standard multivariate normal distribution. Note that $\Omega_{pop} = \omega_{pop}LL^T\omega_{pop}$ where $LL^T$ is the correlation matrix. The prior distributions for the 12 population mean parameters $\theta_{pop}$ are given by uniform distributions with bounds given in *Table 2*. The prior distribution for the 12 population SD parameters is standard half-normal and the prior distribution for the lower Cholesky factor of the correlation matrix $L$ is given by Cholesky LKJ correlation distribution with shape parameter of 2 (*Lewandowski et al., 2009*; *Stan Development Team, 2017*). The distribution of the observed parasitemia measurements is assumed to be a log normal distribution with mean given by the model-simulated values and SD parameter with prior distribution of a half-Cauchy distribution with a location parameter of zero and a scale parameter of 5. The distribution for the observed parasitemia measurements was used to calculate the likelihood function and the M3 method (*Ahn et al., 2008*) was used to penalise the likelihood for data points below the limit of detection for the total parasitemia (10 parasites/mL; *Collins et al., 2018*).

Model fitting was implemented in R (version 3.2.3) (*R Development Core Team, 2017*) and Stan (RStan 2.17.3) (*Stan Development Team, 2017*) using the Hamiltonian Monte Carlo (HMC) optimized by the No-U-Turn Sampler (NUTS) to draw samples from the joint posterior distribution of the parameters including the individual parameters (12 parameters for each volunteer) and population mean parameters (12 hyperparameters). Five chains with different starting points (set by different

random seeds) were implemented and 1000 posterior samples retained from each chain after a burn-in of 1000 iterations (in total 5000 samples were drawn from the joint posterior distribution). The marginal posterior and prior distributions of the population mean and SD parameters are shown in *Figure 1—figure supplements 1* and *2*. The marginal posterior distributions of the individual parameters for all 17 volunteers are shown in *Figure 1—figure supplements 3–14* (using violin plots). For each volunteer, the 5000 sets of individual parameters are used to simulate the gameto-cyte dynamics model and generate 5000 simulated model outputs (e.g., 5000 time series of total parasitemia, asexual parasitemia or gametocytemia). The posterior prediction and 95% prediction interval (PI) are given by the median and quantiles of 2.5% and 97.5% of the 5000 model outputs at each time respectively (see *Figures 1–3* for example).

The estimates of some key biological parameters (*Table 1*) were calculated using the 5000 poste-rior draws of the 12 population mean parameters, that is median and 2.5%- and 97.5%-quantile (95% credible interval). The sexual commitment rate was calculated by $f_{pop} \times 100\%$ ($f_{pop}$ is the popula-tion mean parameter for $f$) and the proportion of committed asexual parasites that survive to become mature gametocytes was calculated by $f_{pop}\left(m_{pop}/\left(m_{pop} + \delta_G pop\right)\right)^4$ where the factor of four arises due to the four sequestered gametocyte stages (I to IV). Circulating gametocyte lifespan was calculated by $1/\delta_{Gm}pop/24$ (the factor of 24 converts hours into days). Gametocyte sequestration time was calculated by $4/m_{pop}/24$ where 4 indicates four sequestered state (stage I to IV) and 24 con-verts hours into days. Parasite multiplication factor is calculated by $r_P pop \exp(-\delta_P pop a_L)\left(1 - f_{pop}\right)$ where the term $\exp(-\delta_P pop a_L)\left(1 - f_{pop}\right)$ gives the fraction of surviving asexual parasites after death and sexual conversion per replication cycle.

The gametocyte dynamics model with parameters given by the median estimates of the popula-tion mean parameters was used to simulate the two scenarios predicting the dependence of human-to-mosquito transmissibility on the sexual commitment rate and gametocyte sequestration time (*Figure 4*).

Final analysis and visualization were performed in MATLAB. All computer codes (R, Stan, MAT-LAB), data and fitting results (CSV and MAT files) and an instruction document (note that reading the document first will make the code much easy to follow) are publicly available at https://doi.org/10.26188/5cde4c26c8201.

## Acknowledgements

We acknowledge useful conversations with David S Khoury, Deborah Cromer and Miles P Davenport (Kirby Institute, UNSW Australia, Sydney, Australia) and assistance in drafting the manuscript from Laura Cascales (QIMR Berghofer, Brisbane, Australia). We thank Jörg J Möhrle, Head of Translation from Medicines for Malaria Venture for his support and permission to use the trial data. This research was supported by use of the Nectar Research Cloud, a collaborative Australian research platform supported by the National Collaborative Research Infrastructure Strategy (NCRIS).

## Additional information

### Funding

| Funder | Grant reference number | Author |
| --- | --- | --- |
| Australian Research Council | Discovery project DP170103076 | James M McCaw |
| National Health and Medical Research Council | 1025319 | Julie A Simpson |
| National Health and Medical Research Council | Centre for Research Excellence 1134989 | Julie A Simpson James McCarthy James M McCaw |
| Australian Research Council | Early Career Researcher Award DE170100785 | Sophie Zaloumis |
| National Health and Medical Research Council | Senior Research Fellowship 1104975 | Julie A Simpson |

| Bill and Melinda Gates Foundation | Thanaporn Wattanakul Joel Tarning |

The funders had no role in study design, data collection and interpretation, or the decision to submit the work for publication.

## Author contributions
Pengxing Cao, Conceptualization, Formal analysis, Visualization, Methodology, Writing—original draft; Katharine A Collins, Data curation, Formal analysis, Writing—review and editing; Sophie Zaloumis, Thanaporn Wattanakul, Joel Tarning, Methodology, Writing—review and editing; Julie A Simpson, Conceptualization, Formal analysis, Writing—review and editing; James McCarthy, Conceptualization, Data curation, Formal analysis, Writing—review and editing; James M McCaw, Conceptualization, Formal analysis, Funding acquisition, Writing—review and editing

## Author ORCIDs
Pengxing Cao ⓘ https://orcid.org/0000-0003-3721-9850
Katharine A Collins ⓘ https://orcid.org/0000-0002-7080-2215
Sophie Zaloumis ⓘ https://orcid.org/0000-0002-8253-8896
Thanaporn Wattanakul ⓘ https://orcid.org/0000-0002-7570-4665
Joel Tarning ⓘ https://orcid.org/0000-0003-4566-4030
Julie A Simpson ⓘ http://orcid.org/0000-0002-2660-2013
James McCarthy ⓘ https://orcid.org/0000-0001-6596-9718
James M McCaw ⓘ https://orcid.org/0000-0002-2452-3098

## Decision letter and Author response
Decision letter https://doi.org/10.7554/eLife.49058.032
Author response https://doi.org/10.7554/eLife.49058.033

# Additional files

## Supplementary files
• Transparent reporting form DOI: https://doi.org/10.7554/eLife.49058.026

## Data availability
All data generated or analysed during this study are included in the manuscript, the supporting information and University of Melbourne's repository (publicly available at https://doi.org/10.26188/5cde4c26c8201).

The following dataset was generated:

| Author(s) | Year | Dataset title | Dataset URL | Database and Identifier |
|---|---|---|---|---|
| Pengxing Cao | 2019 | Data and computer code for model fitting and simulation | https://doi.org/10.26188/5cde4c26c8201 | Melbourne figshare, 10.26188/5cde4c26c8201 |

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

# Appendix 1

DOI: https://doi.org/10.7554/eLife.49058.027

The PK model (see the Materials and methods) describes absorption of administrated drug mass in compartment D and the subsequent kinetics of drug concentration in the central compartment C. The initial conditions for the model simulation were that all the compartments (i.e., D, $T_1$, $T_2$, C, $P_1$ and $P_2$) were zero because PQP was not given until day 7 or 8. When the first dose of 480 mg PQP was given, we set D = 480 and all other compartments to zero. When a second dose of 960 mg was required for some volunteers to treat recrudescence, we again set D to be 960 plus any remaining level of D from the first dose for those volunteers.

The PK data and the PK curves generated using our optimization approach are shown in *Figure 6—figure supplement 1*. The starting values for optimization are $k_T = 1/2.79$, $q_c = 51.4$, $V_c = 804$, $q_1 = 2020$, $V_1 = 3010$, $q_2 = 149$ and $V_2 = 13300$, which are the population-level mean estimates provided by Thanaporn Wattanakul and Joel Tarning (Mahidol-Oxford Tropical Medicine Research Unit, Bangkok). We also assume the parameters are constrained in the optimization procedure by lower bounds of $k_T = 1/3.25$, $q_c = 40.6$, $V_c = 469$, $q_1 = 746$, $V_1 = 2344$, $q_2 = 117$ and $V_2 = 10350$ and upper bounds of $k_T = 1/2.32$, $q_c = 160$, $V_c = 1342$, $q_1 = 5034$, $V_1 = 3986$, $q_2 = 190$ and $V_2 = 17546$. Note that $k_T$ is expressed by the reciprocal of the mean transition time. The lower and upper bounds are the limits of the 95% confidence intervals of the parameter estimate distributions provided by Thanaporn Wattanakul and Joel Tarning except that the upper bound for the clearance rate $q_c$ was increased from 62.6 to 160 such that the PK curves could better capture the fast PQP concentration decay observed for some volunteers. The constraints are necessary due to the limited available PK data which prevents identification of the PK parameters. For Volunteer 202, 301, 302, 307 whose numbers of PK data points are less than the number of parameters in the PK model (such that optimization fails), their 'best-fit' parameter values are the starting values given above with some adjustments on $q_c$ (e.g., 71.4 for 202, 61.4 for 301 and 81.4 for 302) which are required to allow the simulated curves to visually capture the data (as shown in *Figure 6—figure supplement 1* where the predicted PQP concentrations are very close to the observed concentrations).

The PK parameter values obtained from our optimization approach to produce the reasonable PK curves for all 17 volunteers are provided in the *Appendix 1—table 1* below.

**Appendix 1—table 1.** Parameter values used to generate the optimized PK curves for all 17 volunteers. The units of the model parameters are given in the parentheses. The optimized PK curves are shown in *Figure 6—figure supplement 1*.

| PK parameter (unit) | $k_T(h^{-1})$ | $q_c(L/h)$ | $V_c(L)$ | $q_1(L/h)$ | $V_1(L)$ | $q_2(L/h)$ | $v_2(L)$ |
|---|---|---|---|---|---|---|---|
| Volunteer 101 | 1/2.43 | 53.7 | 831 | 2240 | 3958 | 118 | 17177 |
| Volunteer 102 | 1/2.32 | 123.4 | 582 | 849 | 3985 | 163 | 10355 |
| Volunteer 103 | 1/3.07 | 61.3 | 756 | 2686 | 3910 | 127 | 14346 |
| Volunteer 104 | 1/2.32 | 134.7 | 488 | 746 | 3986 | 190 | 15007 |
| Volunteer 105 | 1/3.25 | 160 | 1342 | 5034 | 3986 | 190 | 17546 |
| Volunteer 106 | 1/2.40 | 60.7 | 667 | 2992 | 3043 | 123 | 16374 |
| Volunteer 201 | 1/2.32 | 160 | 469 | 746 | 3986 | 190 | 15557 |
| Volunteer 202 | 1/2.79 | 71.4 | 804 | 2020 | 3010 | 149 | 13300 |
| Volunteer 203 | 1/2.32 | 141.9 | 1243 | 803 | 3973 | 190 | 11825 |

*Appendix 1—table 1 continued on next page*

Appendix 1—table 1 continued

| PK parameter (unit) | $k_T(h^{-1})$ | $q_c$(L/h) | $V_c$(L) | $q_1$(L/h) | $V_1$(L) | $q_2$(L/h) | $v_2$(L) |
|---|---|---|---|---|---|---|---|
| Volunteer 204 | 1/2.32 | 63.3 | 895 | 2169 | 3327 | 165 | 16205 |
| Volunteer 301 | 1/2.79 | 61.4 | 804 | 2020 | 3010 | 149 | 13300 |
| Volunteer 302 | 1/2.79 | 81.4 | 804 | 2020 | 3010 | 149 | 13300 |
| Volunteer 303 | 1/2.45 | 159.4 | 1332 | 779 | 3970 | 190 | 17355 |
| Volunteer 304 | 1/2.48 | 158.9 | 1339 | 762 | 3981 | 190 | 17290 |
| Volunteer 305 | 1/2.53 | 120.9 | 1152 | 1905 | 2979 | 123 | 13896 |
| Volunteer 306 | 1/2.32 | 158.8 | 870 | 761 | 3983 | 190 | 17488 |
| Volunteer 307 | 1/2.79 | 51.4 | 804 | 2020 | 3010 | 149 | 13300 |

DOI: https://doi.org/10.7554/eLife.49058.028

