## [Decision Letter]

Thank you for submitting your article "Modelling the dynamics of *Plasmodium falciparum* gametocytes in humans during malaria infection" for consideration by *eLife*. Your article has been reviewed by Neil Ferguson as the Senior Editor, a Reviewing Editor, and two reviewers. The reviewers have opted to remain anonymous.

The reviewers have discussed the reviews with one another and the Reviewing Editor has drafted this decision to help you prepare a revised submission.

Summary:

In this work, the authors have extended a within-host model of a *P. falciparum* infection to include gametocyte production. The model is calibrated to parasitaemia from 17 volunteers in a controlled human infection study. Although well thought out and well written, a number of concerns remain: (1) several clarifications need to be made in the description of the models; (2) additional justification is needed for the statistical validation of the results; (3) the section on predicting the impact of gametocyte kinetics seems poorly supported.

Essential revisions:

1) Model description

- a and t in the model are never defined (subsection “Study population and measurements”).

- Γ is never defined (subsection “Gametocyte dynamics model”).

- Where do the parameters ranges from Table 2 originate? Why are the death rates so high? For example, the maximum death rate of gametocytes is 0.1/hr, suggesting an average life span of 10 hours.

- Although never stated in this manuscript, it seems that the inoculum from the Collins paper was 2800 parasites. Discuss how the range of 0-10 parasites/mL relates to this.

- Vc, V1 and V2 are never defined (subsection “Pharmacokinetic model of piperaquine (PQP)”).

- How are the initial conditions of the drug model chosen? Why are there two spikes for some patients and only one for others (Figure 6—figure supplement 1)?

- In the fitting of the PK model, was only a single starting set considered? Why is it necessary and valid to increase the upper bound for qc? For different parameters, several patients hit the upper or lower bounds. In particular, Volunteers 105 and 20 hit the bounds on almost all parameters. Could their optimal fit be outside the range?

- Justification for all parameter choices and ranges, specifically when the fitted parameters fall on the bound of these ranges is necessary. If there is a specific biological reason to keep the bounds as they are that should be noted. If there is not a biological reason, the bounds should be widened to show that the parameters fitted, and thus the major conclusions, are not impacted by their choice of bounds.

- Throughout the paper there is discussion of "sexual commitment rate" but reference is always to the percentage commitment. That distinction should be clear and consistent throughout the manuscript. (See Table 1; subsection “Predicting the impact of gametocyte kinetics on human-to-mosquito transmissibility”; Figure 4CD axes labels; Discussion section; Table 2; subsection “Fitting the model to parasitaemia data”; Figure 1—figure supplement 15A axes label; Figure1—figure supplement 1 caption; Figure 1—figure supplement 2 caption; Figure 1—figure supplement 10 caption).

2) Statistical quantification in subsection “Model fitting and validation”, can "very well" be quantified? Can "excellent predictive" be quantified? Also, "very persistent" seems in contradiction of the previous paragraph of discussion of discrepancies. Why is there no discussion in the text of Figure 3? It seems a major point of the paper. How does one validate "visually capture the data" in the caption of Appendix 1?

3) Predicting likelihood of transmission. Although a reference is given for the choice of 10^8^ parasite/mL for newly hospitalized cases, this number differs from the values seen in other important references (Eichner et al.,). For the non-infectious period, the 10^3^ parasites/mL is listed as a value below which there is no transmission. As written in the manuscript it sounds like transmission is likely above this value. Furthermore, Figure 4 is confusing. What is the line in D? The only mention refers to when Gc=10^3^, which is all of the values in D. The scaling (log vs linear) in these figures is confusing. Why use log on the fraction of sexual commitment? Why use linear on the Gc value?

4) Subsection “Predicting the impact of gametocyte kinetics on human-to-mosquito transmissibility” seem to suggest that, when determining the point during the infection at which the patient is hospitalised, total parasite density is a better indicator that asexual parasite density. This statement surprised me: a reference to Saralambda et al., 2010 is provided, but I can't find any mention of this. Have I understood the statement correctly, and is there any evidence of this? This is an important point, as the results presented in Figure 4 depend on the determination of the time of hospitalisation (which presumably is a proxy for the patient becoming febrile). The asexual parasite population is responsible for the rupturing of red blood cells every 48 hours, which is often linked to symptomatic malaria. The statement in subsection “Predicting the impact of gametocyte kinetics on human-to-mosquito transmissibility” seems to be contradicted in the Discussion section, which is a bit confusing.

5) In subsection “Gametocyte dynamics model” the authors state that piperaquine does not kill immature or mature gametocytes, although there is no reference for this. in vitro evidence (S.H. Adjalley et al., 2011) suggests that the drug does have some effect on very young gametocytes (stages I and II). I think it is reasonable to neglect these effects in the model, but less reasonable to state that there aren't any (I appreciate that I'm being a bit fussy here). Neglecting this drug effect may lead to an increased estimate for the death rate of immature gametocytes, but this is just speculation on my part and I'm not suggesting that the model should be re-fitted at this stage.

6) In Figure 2, several of the panels contain data points that look to be below the stated limit of detected (e.g. all data points after day 14 for Volunteer 202). I imagine that these represent zeros (data points for which the parasitaemia was below the limit of detection), this should be stated somewhere if so.

7) I'm a bit confused by the red line in Figure 4D. The caption of the figure states that, "The red curves indicate the cases corresponding to gametocytaemia of 10^3^ parasites/mL". But my understanding of Figure 4D is that at every point (of the 2D surface) gametocytaemia has reached 10^3^ gametocytes / mL. Have I understood this correctly? It could be that the red line on this panel is a contour of constant *t_c_*, which should be clearly stated if this is the case. *The results presented in Figure 4C suggest that newly hospitalised malaria patients are unlikely to be infectious. How do these results compare with clinical trial data? Baseline gametocytaemia is routinely recorded in clinical trials of uncomplicated malaria. It would be interesting to see how the results compare (and why they might be different).

8) One limitation of the study is that the circulation time of mature gametocytes cannot be estimated with any accuracy, due to the lack of patient follow-up. The authors do acknowledge this, but I think the statement (subsection “Estimation of gametocyte dynamics parameters”), "… we found that the circulating gametocyte lifespan… was much longer than that estimated from the neurosyphilis patient data…" is too strong and should be adjusted. In particular, using a constant hazard for gametocyte death (which was not found to provide the best model fit to the neurosyphilis patient data) in the case where an adequate follow up period was not available will overestimate the circulation time of the gametocytes.

9) In Table 2, I wonder if the prior on parameter *f* should be 0-100%, not 0-1%. Figure 1—figure supplement 1 and Figure 1—figure supplement 10 suggests that the relevant marginal posteriors extend beyond 0.01 (here *f* not converted to a percentage). Furthermore, in Table 1 the authors compare the commitment rate to the much higher values obtained elsewhere, particularly in vitro. It would be a curious comparison, if higher values were excluded from the model by using a uniform prior between 0-1%.

10) In Figure 1—figure supplement 1, I was initially confused by the high values for *r_p_*. Toward the end of the Methods section, the authors do explain how to convert this value to a net multiplication rate, which does clarify the matter. This parameter could be called (e.g.) the raw multiplication rate, but I leave this to the authors' discretion.

---

## [Author Response]

Summary:In this work, the authors have extended a within-host model of a P. falciparum infection to include gametocyte production. The model is calibrated to parasitaemia from 17 volunteers in a controlled human infection study. Although well thought out and well written, a number of concerns remain: (1) several clarifications need to be made in the description of the models; (2) additional justification is needed for the statistical validation of the results; (3) the section on predicting the impact of gametocyte kinetics seems poorly supported.

We have made a number of changes to improve the manuscript based on the above comments. We address each of them in detail below and very much appreciate the editors and reviewers’ time for providing the helpful suggestions.

Essential revisions:1) Model description- a and t in the model are never defined (subsection “Study population and measurements”).

The variables a and t were defined in the originally submitted manuscript (Line 350). However, we note that Figure 5, which introduced the notation, was referred to before these definitions. We have revised the caption of Figure 5 in the revised manuscript to ensure that notation is defined on first use:

“The model is comprised of three parts describing three populations of parasites: asexual parasites (Pa,t), sexually committed parasites (PGa,t) and gametocytes (Gt). *P* and *P_G_* are functions of asexual parasite age a and time t. Square compartments in the inner loop represent the asexual parasite population which follows a cycle of maturation and replication every aL hours. Sexual commitment occurs from age as and a fraction of asexual parasites become sexually committed (the bigger square compartments in the outer loop) and eventually enter the development of stage I–V gametocytes (*G_1_–G_5_*). The compartments with a dashed boundary are sequestered to tissues and thus not measurable in a blood smear. The notation for each compartment is consistent with those inthe model equations and is explained in the main text.”

Note that the text in subsection “Study population and measurements” of the originally submitted manuscript has not been changed.

- γ is never defined (subsection “Gametocyte dynamics model”).

*γ* is now defined in the revised manuscript (subsection “Gametocyte dynamics model”):

“…, and γ is the Hill coefficient determining the curvature of the dose-response curve”

- Where do the parameters ranges from Table 2 originate? Why are the death rates so high? For example, the maximum death rate of gametocytes is 0.1/hr, suggesting an average life span of 10 hours.

The parameter ranges shown in Table 2 are for the (Bayesian) prior distributions on the model’s parameters. They should not be interpreted as the biological estimates for the parameters. The biological estimates for the parameters, informed by the VIS data, are given by the Bayesian posterior distributions. The ranges for the prior distributions are conventionally chosen such that they are sufficiently wide to accommodate the estimates in the literature (e.g. for our case some parameter ranges were chosen to be wider than the in vivo values given in Zaloumis et al., 2012) but also confined by biological plausibility (e.g. for our model the lower bounds were all non-negative because of the definitions of the model parameters). Since there is no gold standard for determining how wide the ranges of the (uniform) priors should be, we chose relatively large values for the upper bounds. For example, as the reviewers pointed out, the upper bound of gametocyte death rate was chosen to be 0.1/h (i.e. a mean life span of 10 hours), which is much higher than the maximum of our posterior estimates ~0.009/h (see Figure 1—figure supplement 1). This is actually a clear demonstration of how the different ranges between the prior distribution and the posterior distribution indicate that the (VIS) data holds considerable information on the value for this parameter. We have adjusted the caption of Table 2 in the revised manuscript:

“The table includes the unit, description and prior distribution for each model parameter. For the uniform prior distributions (U), the lower bounds are non-negative based on the definitions of the model parameters and the upper bounds for the prior distributions were chosen to be sufficiently wide in order to accommodate all biologically plausible values from the literature (Zaloumis et al., 2012). We assumed parasites younger than 25h are circulating and thus fix as to be 25h. For 3D7 strain, the asexual replication cycle is approximately 39–45h (based on in vitro estimates (Duffy and Avery, 2017) and personal communication (JS McCarthy, personal communication, May 2019)) and we fix aL to be 42h.”

- Although never stated in this manuscript, it seems that the inoculum from the Collins paper was 2800 parasites. Discuss how the range of 0-10 parasites/mL relates to this.

Once again, we note that our range of 0–10 is for the prior distribution, not an estimate for the range of the actual value. Furthermore, there is no simple conversion from the inoculum of 2800 infected red blood cells to the initial parasite density due to a number of factors, such as blood volume, stochastic variation in the proportion of viable parasites, heterogeneity of the parasites in the blood, inter-individual variation in parasite growth rates etc. We chose an upper bound on the prior of 10 because the initial total parasite load was below the detection limit of 10 parasites/mL. Note that even on day 4 where the total parasitaemia should have been at least two orders of magnitude more than the initial parasitaemia, none of the volunteers had a parasitaemia level > 100 parasites/mL, suggesting that the initial parasite load must be far less than 10. As above, the credible range of the posterior distribution is what constitutes the biological contribution of our analysis. The posterior for the initial parasite load *P_init_* was found to be 0.0232–0.2948, as seen in Figure 1—figure supplement 1, indicating that the chosen upper bound for the prior was both appropriate and had an absolutely negligible effect on the posterior distribution.

- Vc, V1 and V2 are never defined (subsection “Pharmacokinetic model of piperaquine (PQP)”).

Thank you for identifying this oversight. We have explained them in subsection “Pharmacokinetic model of piperaquine (PQP)” in the revised manuscript:

“…, where kT and q’s are rate constants as shown in Figure 6 and Vc, V1 and V2 are the volume of distribution for the central compartment (in which PQP concentration is *C*), peripheral compartment 1 (in which PQP concentration is *P*_1_) and peripheral compartment 2 (in which PQP concentration is *P*_2_) respectively.”

- How are the initial conditions of the drug model chosen?

Thank you for identifying this oversight. The initial conditions for PK model simulation were that all the compartments (i.e. *D, T_1_, T_2_, C, P_1_*and *P_2_*) were zeroes because PQP was not given until day 7 or 8. When the first dose of 480mg PQP was given, we set *D* = 480 and zero for other compartments. When a second dose of 960mg was required for some volunteers to treat recrudescence, we set *D* to be 960 plus any remaining level of *D* from the first dose for those volunteers. Note our PK model (as described in the Section “Pharmacokinetic model of piperaquine (PQP)”) explicitly models the absorption of administrated drug mass in compartment *D*, and how that drives drug concentration in the central compartment *C*, and so the second dose is modelled as a step change in *D*, not a step change in *C*. We now describe how the initial conditions are determined in the Appendix 1:

“The PK model (see the Materials and methods section) describes absorption of administrated drug mass in compartment *D* and the subsequent kinetics of drug concentration in the central compartment *C.* The initial conditions for the model simulation were that all the compartments (i.e. *D, T_1_, T_2_, C, P_1_* and *P_2_*) were zero because PQP was not given until day 7 or 8. When the first dose of 480mg PQP was given, we set *D* = 480 and all other compartments to zero. When a second dose of 960mg was required for some volunteers to treat recrudescence, we again set *D* to be 960 plus any remaining level of *D* from the first dose for those volunteers.”

and in the main text in the revised manuscript (subsection “Fitting the model to parasitaemia data”):

“The details of the initial conditions, starting point and constraints for the PK curve optimisation procedure are provided in Appendix 1.”

Why are there two spikes for some patients and only one for others (Figure 1—figure supplement 1)?

A second dose of PQP was given to the volunteers showing the recrudescence. While this was mentioned in the first section of the Materials and methods section in the originally submitted manuscript, we now also mention it in the caption of Figure 6—figure supplement 1:

“Some volunteers have two peaks of PQP concentrations because they had recrudescent asexual parasitaemia (see Figure 1 in the main text) and were treated with a second dose of 960mg PQP.”

- In the fitting of the PK model, was only a single starting set considered? Why is it necessary and valid to increase the upper bound for qc? For different parameters, several patients hit the upper or lower bounds. In particular, Volunteers 105 and 20 hit the bounds on almost all parameters. Could their optimal fit be outside the range?

These questions all concern the PK parameters used for each volunteer to generate PQP concentration data for our gametocyte dynamics model fitting. The Volunteer Infection Study (VIS) on which our paper is based was not designed to accurately determine the PK parameters for each trial participant. Nonetheless, we did have some PK data for each volunteer. For our gametocyte dynamics model fitting, our primary aim was to determine an appropriate PK curve for each volunteer (rather than a unique estimate of their PK parameters) based on the limited data. That is, we have taken a sequential PK-PD approach, where predicted PQP concentration (i.e. *C(t*)) for each individual is included as a variable in the PD model. To achieve this requirement, we drew on an analysis of rich PK data for PQP from a different VIS, where the PQP’s PK model parameter estimates were determined by our collaborators at Mahidol-Oxford Tropical Medicine Research Unit (Bangkok), Thanaporn Wattanakul and Joel Tarning.

The approach we took, as described (and now further clarified) in the Section “Pharmacokinetic model of piperaquine (PQP)” and the Appendix 1, was to use the MATLAB built-in least-squares optimiser *lsqcurvefit* (with the default setting) to optimise the PK curve for each volunteer in the VIS study. We applied the optimiser to each volunteer’s (limited) PQP data, using the parameter estimates provided by Thanaporn Wattanakul and Joel Tarning as initial values. We applied some further model parameter constraints as specified in the Appendix 1. This approach provided us with the required data-informed PK curve for each volunteer in the VIS, sufficient for our sequential PK-PD study focussing on the asexual and sexual parasite dynamics. Of note, Volunteers 202, 301, 302 or 307 had fewer PK data points than PK model parameters, preventing application of this optimisation procedure. For these volunteers, their predicted PQP PK curve was derived using the population-level mean parameter estimates from Wattanakul and Tarning’s analysis. We have modified the manuscript to clarify the method for producing appropriate PK curves (subsection “Pharmacokinetic model of piperaquine (PQP)”):

“Under the sequential pharmacokinetic-pharmacodynamic (PK-PD) approach we have taken, a PQP concentration curve (*C(t*)) for each volunteer is a required input into the gametocyte dynamics model. The VIS, with its limited sampling of PQP for each volunteer, was not designed to provide this PQP concentration curve directly, so we used a PK model, informed by data from a previous VIS with rich sampling. We drew on an analysis of that previous VIS by Thanaporn Wattanakul and Joel Tarning (unpublished data and estimates). Their analysis provides population-level PQP PK model parameter estimates.

We used MATLAB’s (version 2016b; The MathWorks, Natick, MA) built-in least-squares optimiser *lsqcurvefit* (with the default setting) to optimise the PK curve for each volunteer in the VIS study. We applied the optimiser to each volunteer’s (limited) PQP data, using the parameter estimates provided by Thanaporn Wattanakul and Joel Tarning as initial values. We applied some further model parameter constraints as specified in Appendix 1. This approach provided us with a data-informed PK curve for each volunteer in the VIS, sufficient for our primary purpose of studying the asexual and sexual parasite dynamics. Of note, Volunteers 202, 301, 302 or 307 had fewer PK data points than PK model parameters, preventing application of this optimisation procedure. For these volunteers, their predicted PQP PK curve was derived using the population-level mean PK parameter from Wattanakul and Tarning’s analysis. The MATLAB code (with detailed comments) is publicly available at https://doi.org/10.26188/5cde4c26c8201. The details of the initial conditions, starting point and constraints for the PK curve optimisation procedure are provided in Appendix 1. The optimised PK curves and associated parameter values for all volunteers are provided in Figure 6-figure-supplement 1 and Appendix 1.”

We increased qc in order to generate reasonable PK curves. Because some volunteers (e.g. 102, 104, 105, 201, 306, etc.) exhibited a fast drug decay, it was necessary to relax the upper bound of qc to improve the fit to the (limited) available data. We have explained this in Appendix 1:

“… except that the upper bound for the clearance rate qc was increased from 62.6 to 160 such that the PK curves could better capture the fast PQP concentration decay observed for some volunteers.”

- Justification for all parameter choices and ranges, specifically when the fitted parameters fall on the bound of these ranges is necessary. If there is a specific biological reason to keep the bounds as they are that should be noted. If there is not a biological reason, the bounds should be widened to show that the parameters fitted, and thus the major conclusions, are not impacted by their choice of bounds.

Our previous response to the questions of parameter estimation – and that what we require for our analysis of the VIS data is reasonable PK curves, not PK parameter estimates – has partly addressed this concern. For the PK model, the bounds were chosen to be the 95% confidence interval of the PK parameter estimates provided by Thanaporn Wattanakul and Joel Tarning, with some relaxation on the upper bound of the clearance rate 𝑞𝑐 as mentioned above.

We have adjusted the text in Appendix 1:

“The lower and upper bounds are the limits of the 95% confidence intervals of the parameter estimate distributions provided by Thanaporn Wattanakul and Joel Tarning except that the upper bound for the clearance rate 𝑞𝑐 was increased from 62.6 to 160 such that the PK curves could better capture the fast PQP concentration decay observed for some volunteers. The constraints are necessary due to the limited available PK data which prevents identification of the PK parameters.”

- Throughout the paper there is discussion of "sexual commitment rate" but reference is always to the percentage commitment. That distinction should be clear and consistent throughout the manuscript. (See Table 1; subsection “Predicting the impact of gametocyte kinetics on human-to-mosquito transmissibility”; Figure 4CD axes labels; Discussion section; Table 2; subsection “Fitting the model to parasitaemia data”; Figure 1—figure supplement 15A axes label; Figure1—figure supplement 1 caption; Figure 1—figure supplement 2 caption; Figure 1—figure supplement 10 caption).

Thank you for identifying this ambiguity in nomenclature. By rate, we refer to a “per replication cycle rate”. The revised manuscript uses a new terminology: the sexual commitment rate is now consistently expressed as “percentage/asexual replication cycle” (e.g. 0.54%/asexual replication cycle). The parameter *f* indicates the fraction and thus lies between 0 and 1. The conversion between the sexual commitment rate and parameter *f* is “sexual commitment rate = *f*×100%”. We have corrected any ambiguous expressions and provided the conversion method (subsection “Fitting the model to parasitaemia data”) in the revised manuscript.

2) Statistical quantification in subsection “Model fitting and validation”, can "very well" be quantified? Can "excellent predictive" be quantified? How does one validate "visually capture the data" in the caption of Supplementary file 1?

In the Bayesian analysis literature, much of model validation is qualitative, based on a combination of measures and analysis including the posterior predictive check. In our context, “very well” or "excellent predictive capability" means the predicted parasitaemia (i.e. the median and 95% prediction interval) is able to accurately capture the detailed dynamics/trajectory of the data. This was primarily judged by visual check rather than based on quantities measuring the goodness of fit, such as information criteria, which are often less meaningful (i.e. no better than visual check) unless used for model comparison purposes. We have revised the manuscript (Introduction):

“The results show that the predicted total parasitaemia (median and 95% PI) is able to accurately capture the trends of the data through the (visual) posterior predictive check.”

and (subsection “Model fitting and validation”):

“For the majority of asexual parasitaemia data the model predictions (median and 95% PI) can faithfully capture the trends of the data (Figure 2)”.

For determining patient-specific PQP concentrations at the times when parasitaemia was measured (which is an essential step of our sequential PK-PD approach), we aimed to generate PK curves which are visually close to the limited PK data (note that the difficulty of fitting to the limited PK data was explained in detail in our response to comment (1)), and one can validate the quality of the predicted PQP concentrations by looking at Figure 6—figure supplement 1 where both the predicted PQP profiles and available data are provided for all the volunteers. We have replaced “visually capture the data” by “visually capture the data (as shown in Figure 6—figure supplement 1 where the predicted PQP concentrations are very close to the observed concentrations)” in the Appendix 1 to minimise the ambiguity.

In the subsection “Model fitting and validation”, "very persistent" seems in contradiction of the previous paragraph of discussion of discrepancies. Why is there no discussion in the text of Figure 3? It seems a major point of the paper.

We begin by noting that presumably the reviewers are asking about the use of the phrase “very consistent” rather than (as typed) “very persistent”. Figure 3 was discussed, albeit briefly, in subsection “Model fitting and validation” of the originally submitted manuscript (which is the text being queried by the reviewers).

In the originally submitted manuscript, Figure 3 shows that the predicted gametocytaemia for all volunteers is very close to the measured data for those volunteers, hence our description as “very consistent”. Note that we acknowledged that the model’s predictions of the asexual parasitaemia (not the gametocytaemia) for some volunteers were less consistent (subsection “Model fitting and validation” which refers to Figure 2).

We have modified the manuscript to ensure that these very important results in Figure 3 are clearly emphasised for the reader. We have modified the discussion of Figure 3 and highlighted it as a new paragraph in the revised manuscript (subsection “Model fitting and validation”):

“Figure 3 shows the data and model predictions for the gametocytaemia. Despite some discrepant observations for asexual parasitaemia in Figure 2, we found that the model predictions of gametocytaemia were able to capture the trends and levels of the gametocytaemia data for all 17 volunteers.”

3) Predicting likelihood of transmission. Although a reference is given for the choice of 10^8^ parasite/mL for newly hospitalized cases, this number differs from the values seen in other important references (Eichner et al.,). For the non-infectious period, the 10^3^ parasites/mL is listed as a value below which there is no transmission. As written in the manuscript it sounds like transmission is likely above this value. Furthermore, Figure 4 is confusing. What is the line in D? The only mention refers to when Gc=10^3^, which is all of the values in D. The scaling (log vs linear) in these figures is confusing. Why use log on the fraction of sexual commitment? Why use linear on the Gc value?

We used a unit of “parasites per milli litre” while Eichner et al. used “parasites per micro litre”. A unit conversion of 10^8^ parasites/mL gives 10^5^ parasites/uL which matches the level shown in Eichner et al.

For the threshold of 10^3^, we have adjusted subsection “Predicting the impact of gametocyte kinetics on human-to-mosquito transmissibility” in the revised manuscript:

“…, which is a threshold below which human-to-mosquito transmission was not observed (Collins et al., 2018).”

We have provided the definition of the red curve in Figure 4D in the revised manuscript (subsection “Predicting the impact of gametocyte kinetics on human-to-mosquito transmissibility”):

“The red curve in C is the level curve for *G_c_ =* 10^3^ parasites/mL. The red curve in D is the level curve for *t_c_*= 13.42 days which is the non-infectious period obtained by model simulation using the posterior estimates of the population mean parameters.”

We used a log scale for the sexual commitment rate because the plausible range of the rate spanned a few orders of magnitude (i.e. from order 0.1% to 10%; our in vivo estimate was about 0.5% while the in vitro estimate is about 10%). For *G_c_*, although parasitaemia is typically shown on a log scale, the variation in *G_c_*was roughly linear (see the colour bar in Figure 4C) and therefore we used a linear scale for *G_c._*. Based on these reviewers’ queries, we have considered alternative plots but have decided that our current visualisations are the most appropriate.

4) Subsection “Predicting the impact of gametocyte kinetics on human-to-mosquito transmissibility” seem to suggest that, when determining the point during the infection at which the patient is hospitalised, total parasite density is a better indicator that asexual parasite density. This statement surprised me: a reference to Saralambda et al., [2010] is provided, but I can't find any mention of this. Have I understood the statement correctly, and is there any evidence of this? This is an important point, as the results presented in Figure 4 depend on the determination of the time of hospitalisation (which presumably is a proxy for the patient becoming febrile). The asexual parasite population is responsible for the rupturing of red blood cells every 48 hours, which is often linked to symptomatic malaria. The statement in subsection “Predicting the impact of gametocyte kinetics on human-to-mosquito transmissibility” seems to be contradicted in the Discussion section, which is a bit confusing.

Thank you for pointing out the confusing statement (Subsection “Predicting the impact of gametocyte kinetics on human-to-mosquito transmissibility”) and a typographical error in the Discussion section. First of all, “asexual parasitaemia” should be “total parasitaemia”, as the total parasitaemia was consistently used in the simulation of the clinical scenario and in the Results section. We have changed “asexual” to “total” in the revised manuscript (Discussion section).

The reference to Saralamba et al., 2010 was to support the choice of 10^8^ parasites/mL as a realistic value for the total parasitaemia of newly hospitalised patients. Since the parasitaemia data in Saralamba et al., 2010 was measured under microscope where both asexual and sexual rings were counted, we chose total parasitaemia to simulate the clinical scenario. We have modified the statement (Lines 199-200 in the originally submitted manuscript) in the revised manuscript to be (subsection “Predicting the impact of gametocyte kinetics on human-to-mosquito transmissibility”):

“We further assumed that patients would seek hospital admission when their total parasitaemia reached approximately 10^8^ parasites/mL. This choice was based on the microscopic measurements of the total parasitaemia (i.e. asexual and sexual parasites) from a study of Cambodia and Thailand hospitalised malaria patients (Saralamba et al., 2010).”

5) In subsection “Gametocyte dynamics model” the authors state that piperaquine does not kill immature or mature gametocytes, although there is no reference for this. in vitro evidence (S.H. Adjalley et al., 2011) suggests that the drug does have some effect on very young gametocytes (stages I and II). I think it is reasonable to neglect these effects in the model, but less reasonable to state that there aren't any (I appreciate that I'm being a bit fussy here). Neglecting this drug effect may lead to an increased estimate for the death rate of immature gametocytes, but this is just speculation on my part and I'm not suggesting that the model should be re-fitted at this stage.

We agree that our statement was inaccurate and have modified the manuscript (subsection “Gametocyte dynamics model”):

“Note that we assumed in our model that PQP does not kill gametocytes. Our assumption was based on evidence from both in vitro and in vivo experiments that suggests that PQP has little activity against sexually committed parasites and gametocytes (Bolscher et al., 2014; Collins et al., 2018; Pasay et al., 2016), although we note there is some evidence that PQP may have activity against early-stage I/II gametocytes (Adjalley et al., 2011).”

6) In Figure 2, several of the panels contain data points that look to be below the stated limit of detected (e.g. all data points after day 14 for Volunteer 202). I imagine that these represent zeros (data points for which the parasitaemia was below the limit of detection), this should be stated somewhere if so.

This interpretation is correct in that the points are those with a read-out of below the limit of detection, but we cannot ever say that the values are actually zero (as in no parasites at all). The data points placed at one parasite/mL (or zero on a log scale) represent the measurements for which no parasites were detected. We now clearly state this in the caption of Figure 2 in the revised manuscript:

“The data points with one parasite/mL (i.e. those points which lie on the dotted line) indicate measurements for which no parasites were detected.”

7) I'm a bit confused by the red line in Figure 4D. The caption of the figure states that, "The red curves indicate the cases corresponding to gametocytaemia of 10^3^ parasites/mL". But my understanding of Figure 4D is that at every point (of the 2D surface) gametocytaemia has reached 10^3^ gametocytes / mL. Have I understood this correctly? It could be that the red line on this panel is a contour of constant t_c_, which should be clearly stated if this is the case.

Thank you for raising this issue of interpretation, which was also raised in comment (3) above. We have addressed it in the revised manuscript (Figure 4 caption):

“The red curve in C is the level curve for *G_c_ =* 10^3^ parasites/mL. The red curve in D is the level curve for *t_c_*= 13.42 days which is the non-infectious period obtained by model simulation using the posterior estimates of the population mean parameters.”

The results presented in Figure 4C suggest that newly hospitalised malaria patients are unlikely to be infectious. How do these results compare with clinical trial data? Baseline gametocytaemia is routinely recorded in clinical trials of uncomplicated malaria. It would be interesting to see how the results compare (and why they might be different).

This is a good suggestion. Patients from field clinical observations of uncomplicated malaria may carry more gametocytes than what our model predicts at the time of presentation, for example one study from the TRACII clinical trial reported a range of 16–5120 gametocytes/µL, much higher than our prediction of below 1 gametocytes/µL (or 10^3^ gametocytes/mL) (Pluijm et al., 2019). One explanation for the difference would be that our model predicted a very fast rise in parasitaemia to 10^8^ parasites/mL while patients in clinical trials with naturally occurring malaria may take a longer time to reach such a high parasitaemia level due to immunity. Immunity was, of course, not considered in our model due to the design of our Volunteer Infection Studies where only malaria naïve volunteers are recruited. Thus, we have added the following in the revised manuscript (subsection “Predicting the impact of gametocyte kinetics on human-to-mosquito transmissibility”) to address this comment:

“Note that patients from clinical observations of uncomplicated malaria in endemic settings may have higher gametocyte counts at the time of presentation than what our model predicts. For example one study from the TRACII clinical trial reported a range of 16–5120 gametocytes/µL, which is much higher than our prediction of below 1 gametocytes/µL (or 10^3^ gametocytes/mL) (van der Pluijm et al., 2019). One plausible explanation for the difference is that our model predicted a very fast rise in total parasitaemia to 10^8^ parasites/mL while the rise in parasitaemia among patients in endemic settings may be slower due to the effect of immunity on the parasite multiplication. Immunity was not considered in our model due to the design of our VIS where only malaria naïve volunteers were recruited.”

8) One limitation of the study is that the circulation time of mature gametocytes cannot be estimated with any accuracy, due to the lack of patient follow-up. The authors do acknowledge this, but I think the statement (subsection “Estimation of gametocyte dynamics parameters”), "… we found that the circulating gametocyte lifespan… was much longer than that estimated from the neurosyphilis patient data…" is too strong and should be adjusted. In particular, using a constant hazard for gametocyte death (which was not found to provide the best model fit to the neurosyphilis patient data) in the case where an adequate follow up period was not available will overestimate the circulation time of the gametocytes.

We fully accept this critique of our original comment. Noting that our comments in the Discussion section emphasise that we cannot make a meaningful estimate of this parameter based on the VIS data, we have decided to simply delete this statement from the revised manuscript.

9) In Table 2, I wonder if the prior on parameter f should be 0-100%, not 0-1%. Figure 1—figure supplement 1 and Figure 1—figure 10 suggests that the relevant marginal posteriors extend beyond 0.01 (here f not converted to a percentage). Furthermore, in Table 1 the authors compare the commitment rate to the much higher values obtained elsewhere, particularly in vitro. It would be a curious comparison, if higher values were excluded from the model by using a uniform prior between 0-1%.

Yes, the reviewers understood correctly that the prior on parameter *f* was defined to be 0-1 (i.e. 0-100%), which allowed the range of posterior distribution to extend beyond 0.01 (i.e. 1%) shown in Figure 1—figure supplement 1 and Figure 1—figure supplement 10. In order to avoid any confusion between the parameter *f* (which indicates the fraction) and the sexual commitment rate (which was expressed as percentage), as for our response to an earlier comment, we have made adjustments throughout the manuscript.

10) In Figure 1—figure supplement 1, I was initially confused by the high values for r_p_. Toward the end of the Methods section, the authors do explain how to convert this value to a net multiplication rate, which does clarify the matter. This parameter could be called (e.g.) the raw multiplication rate, but I leave this to the authors' discretion.

We appreciate that the reviewers have raised this matter which we believe many readers familiar with the details of malaria kinetics may also note. We have added the following text immediately following the introduction of the parasite replication rate rP in the revised manuscript (subsection “Gametocyte dynamics model”):

“Note that we distinguish the parasite replication rate rP from the so-called parasite multiplication factor, the latter of which is a “net multiplication rate” quantified by the (per cycle) increase in parasite numbers due to replication (rP) and the decrease in parasite numbers due to death or sexual commitment.”